# Neural Monge map estimation and its applications

**Jiaojiao Fan**[*]                                                                                    *jiaojiaofan@gatech.edu*
*Georgia Institute of Technology*

**Shu Liu**[*]                                                                                          *shuliu@math.ucla.edu*
*University of California, Los Angeles*

**Shaojun Ma**                                                                                        *shaojunma@gatech.edu*
*Georgia Institute of Technology*

**Hao-Min Zhou**                                                                                      *hmzhou@math.gatech.edu*
*Georgia Institute of Technology*

**Yongxin Chen**                                                                                      *yongchen@gatech.edu*
*Georgia Institute of Technology*

**Reviewed on OpenReview:** *https://openreview.net/forum?id=2mZSlQscj3*

## Abstract

Monge map refers to the optimal transport map between two probability distributions and provides a principled approach to transform one distribution to another. Neural network-based optimal transport map solver has gained great attention in recent years. Along this line, we present a scalable algorithm for computing the neural Monge map between two probability distributions. Our algorithm is based on a weak form of the optimal transport problem, thus it only requires samples from the marginals instead of their analytic expressions, and can be applied in large-scale settings. Furthermore, using the duality gap we prove rigorously *a posteriori* error analysis for the method. Our algorithm is suitable for general cost functions, compared with other existing methods for estimating Monge maps using samples, which are usually for quadratic costs. The performance of our algorithms is demonstrated through a series of experiments with both synthetic and realistic data, including text-to-image generation, class-preserving map, and image inpainting tasks.

## 1 Introduction

The past decade has witnessed great success of optimal transport (OT) (Villani, 2008) based applications in machine learning community (Arjovsky et al., 2017; Krishnan & Martínez, 2018; Li et al., 2019; Inoue et al., 2020; Ma et al., 2020; Fan et al., 2020; Haasler et al., 2020; Alvarez-Melis & Fusi, 2020; Alvarez-Melis et al., 2021; Bunne et al., 2021; Mokrov et al., 2021; Bunne et al., 2022; Yang & Uhler, 2018; Fan et al., 2022a; Fan & Alvarez-Melis, 2022; Nguyen & Ho, 2022; Zhu et al., 2021). The Wasserstein distance induced by OT is widely used to evaluate the discrepancy between distributions thanks to its weak continuity and robustness. In this work, given any two probability distributions $\rho_a$ and $\rho_b$ defined on $\mathbb{R}^n$ and $\mathbb{R}^m$, we consider the **Monge problem**

$$C_{\text{Monge}}(\rho_a, \rho_b) \triangleq \min_{\substack{T:\mathbb{R}^n \to \mathbb{R}^m, \\ T_\sharp \rho_a = \rho_b}} \int_{\mathbb{R}^n} c(x, T(x)) \rho_a(x) \, dx. \tag{1}$$

Here $c(x, y)$ denotes the cost of transporting from $x$ to $y$ and $T$ is the transport map. We define the pushforward of distribution $\rho_a$ by $T$ as $T_\sharp \rho_a(E) = \rho_a(T^{-1}(E))$ for any measurable set $E \subset \mathbb{R}^m$. The Monge

---

[*]Equal contribution.

problem seeks the cost-minimizing transport plan $T_*$ from $\rho_a$ to $\rho_b$. The optimal $T_*$ is also known as the **Monge map** of (1).

In this work, we propose a scalable algorithm for estimating the Wasserstein distance as well as the optimal map in continuous spaces without introducing any regularization terms. Particularly, we apply the Lagrangian multiplier directly to the Monge problem and obtain a sup-inf problem. **Our contribution** is summarized as follows: 1) We develop a neural network based algorithm to compute the optimal transport map associated with general transport costs between two distributions; 2) Our method is capable of computing OT problems between distributions with different dimensions. 3) We provide a rigorous *a posteriori* error analysis of the algorithm based on duality gaps; 4) We demonstrate its performance and its scaling properties in truly high dimensional settings through various experiments.

Just like other computational OT methods, our method does not require paired data for learning the transport map. In real-life applications, the acquirement and maintenance of large-scale paired data are laborious. We will show that our proposed algorithm can be applied to multiple cutting-edge tasks, where the dominant methods still require paired data. In particular, our examples include text-to-image generation and image inpainting, which confirm that our algorithm can achieve competitive results even without paired data.

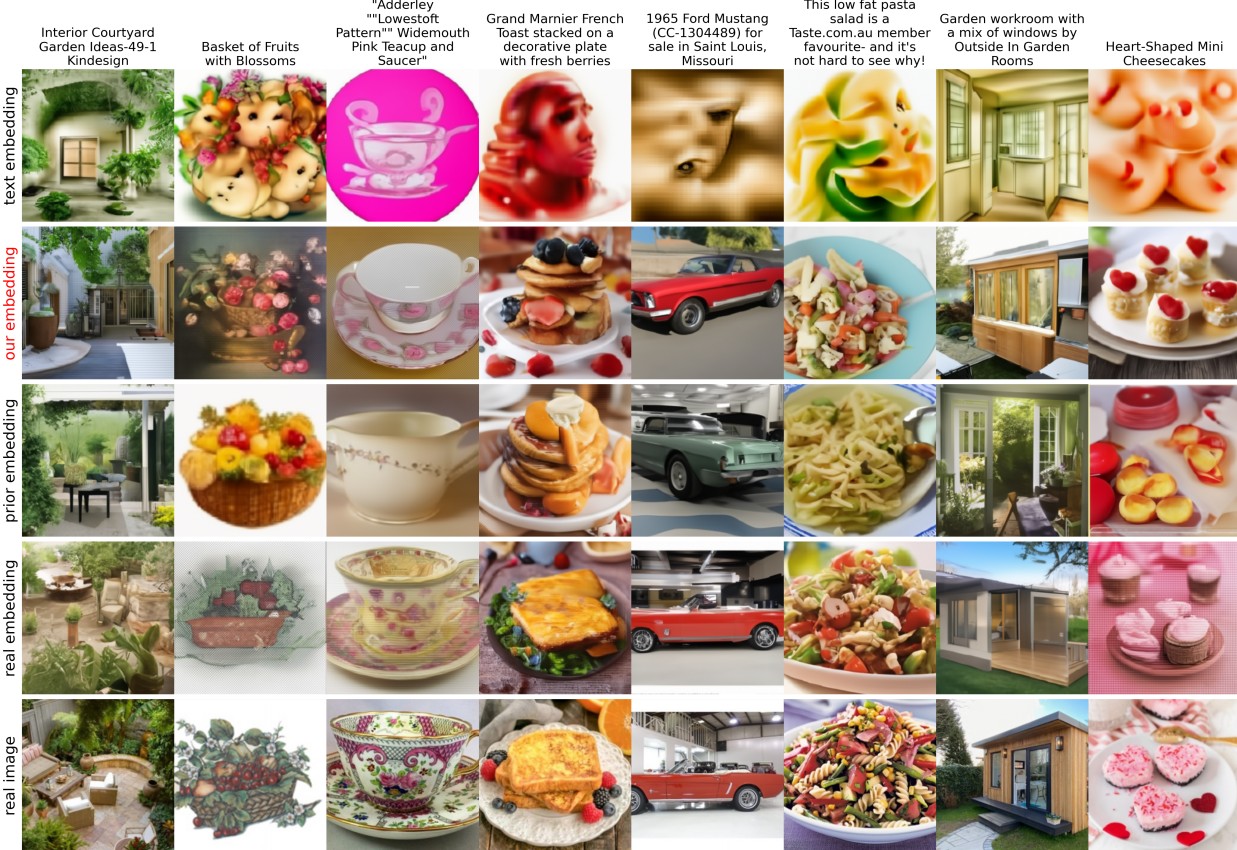

Figure 1: Random image samples with size $224 \times 224$ on Laion art testing prompts. Except for the last row of real images, we show the outputs of the same decoder with different conditions. The condition of decoder includes a text encoding, which we keep unchanged across rows, and an image embedding. In the first row, we feed the decoder the text embedding, and the generated images are unrealistic because the image and the text embeddings are not interchangeable. In the second row, we pass the embedding mapped by our transport map, which is trained on unpaired data. As a baseline method, we pass an image embedding generated by the diffusion prior in DALL·E2-Laion, which is trained on paired data. To explicitly show the decoder's recovery ability, we pass the real image embedding in the fourth row.

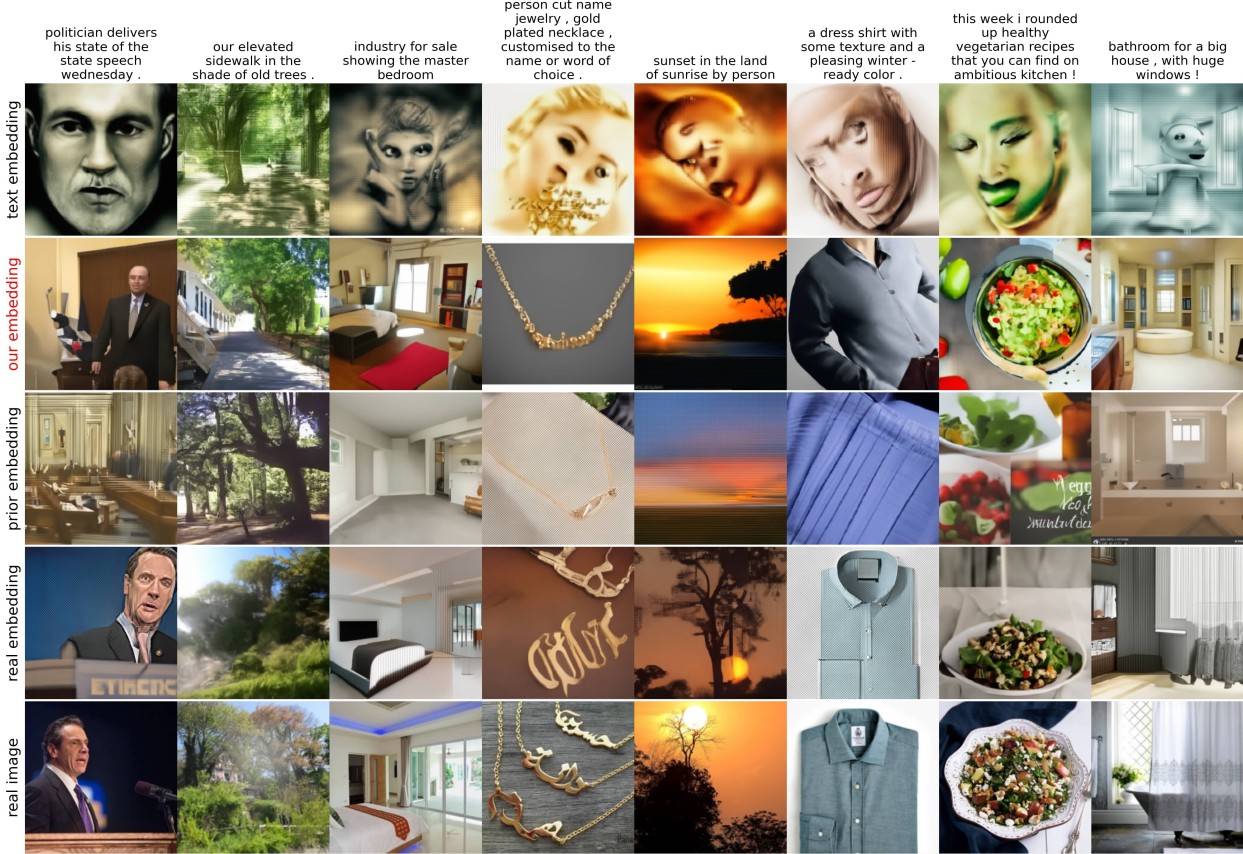

Figure 2: Random image samples on Conceptual Captions 3M prompts. The meaning of each row is the same as in Figure 1.

## 2 A brief background of OT problems

Consider $\mathcal{X}$ as a closed subset of $\mathbb{R}^n$ and $\mathcal{Y}$ as an arbitrary Polish space[1]. Suppose the cost function $c$ is defined on $\mathbb{R}^n \times \mathcal{Y}$. We denote $\rho_a, \rho_b$ as Borel probability distributions defined on $\mathcal{X}, \mathcal{Y}$ respectively. Then the general OT problem, which can be treated as a relaxation of Monge problem (1), can be formulated as

$$C(\rho_a, \rho_b) := \inf_{\pi \in \Pi(\rho_a, \rho_b)} \iint_{\mathcal{X} \times \mathcal{Y}} c(x, y) d\pi(x, y), \tag{2}$$

where we define $\Pi(\rho_a, \rho_b)$ as the set of joint Borel probability distributions on $\mathcal{X} \times \mathcal{Y}$ with marginals equal to $\rho_a$ and $\rho_b$. Once $c$ is continuous and satisfies (15) and $\rho_a$ is atomless, then optimal values of Monge problem and Kantorovich problem are equal to each other (Pratelli, 2007), i.e., $C_{\mathrm{Monge}}(\rho_a, \rho_b) = C(\rho_a, \rho_b)$.

The Kantorovich dual formulation (Villani, 2008, Ch. 5) of the primal OT problem (2) is

$$\sup_{\phi, \psi} \left\{ \int_{\mathcal{Y}} \phi(y) \rho_b(y) \, dy - \int_{\mathcal{X}} \psi(x) \rho_a(x) \, dx \right\}, \tag{3}$$

where the maximization is over all $\psi \in C_b(\rho_a)$, $\phi \in C_b(\rho_b)$ that satisfy $\phi(y) - \psi(x) \leq c(x, y)$ for any $x \in \mathcal{X}, y \in \mathcal{Y}$. For further discussions on the dual problem (3), as well as its equivalence to the primal OT problem (2), we refer the reader to Appendix A. In general, the optimal solution $\pi_*$ of (2) can be treated as a stochastic transport plan, i.e., we are allowed to break a single particle into pieces and then transport each

---

[1]A topological space is Polish space if it is a separable, complete, metric space. Euclidean space $\mathbb{R}^m$ and its closed subsets are Polish spaces.

piece to certain positions according to the plan $\pi_*$. However, in this study, we mainly focus on computing the deterministic optimal map $T_*$ for the Monge problem.

The following theorem states the existence and uniqueness of the optimal solution to the Monge problem. It also reveals the relation between the optimal map $T_*$ and the optimal transport plan $\pi_*$. This theorem is taken from Chapter 10 of (Villani, 2008). A more complete statement can be found in Appendix A.

**Theorem 1.** *(Informal, existence and uniqueness of Monge map) Assume the boundary $\partial \mathcal{X}$ of $\mathcal{X}$ has dimension no larger than $n-1$. Suppose the cost $c$ satisfies* (15), (16), (17), (18), *and assume that $\rho_a$ and $\rho_b$ are compactly supported and $\rho_a$ is absolute continuous with respect to the Lebesgue measure on $\mathbb{R}^n$. Then there exists a unique-in-law[2] transport map $T_*$ solving Monge problem* (1). *Furthermore, $(Id, T_*)_{\sharp}\rho_a$[3] is an optimal solution to the OT problem* (2).

## 3 Proposed method

In order to formulate a tractable algorithm for the general Monge problem (1), we first notice that (1) is a constrained optimization problem. Thus, we introduce a Lagrange multiplier $f$ for the constraint $T_{\sharp}\rho_a = \rho_b$ and reformulate (1) as a saddle point problem

$$\sup_{f \in C_b(\mathcal{Y})} \inf_{T \in \mathcal{M}(\mathcal{X}, \mathcal{Y})} \mathcal{L}(T, f). \tag{4}$$

Here we denote $C_b(\mathcal{Y})$ as the space of bounded continuous functions on $\mathcal{Y}$; and $\mathcal{M}(\mathcal{X}, \mathcal{Y})$ as the space of measurable map $T : \mathcal{X} \to \mathcal{Y}$. In the following discussion, if not specified, we always treat the optimization space of $f$ and $T$ as $C_b(\mathcal{Y}), \mathcal{M}(\mathcal{X}, \mathcal{Y})$. Furthermore, we define $\mathcal{L}(T, f)$ as

$$\int_{\mathcal{X}} c(x, T(x))\rho_a dx + \int_{\mathcal{Y}} f(y)(\rho_b - T_{\sharp}\rho_a)dy = \int_{\mathcal{X}} [c(x, T(x)) - f(T(x))]\, \rho_a dx + \int_{\mathcal{Y}} f(y)\rho_b dy. \tag{5}$$

It is worth mentioning that if we optimize the inner problem of (4), our proposed problem (4) is also closely tied to the Kantorovich dual problem (20) of primal OT problem. Such observation will be useful on establishing several theoretical results for our method.

The following theorem reveals the equivalence between the existence of solutions to primal problem (1) as well as dual problem (3) and the existence of saddle point (see Definition 3 of Appendix B).

**Theorem 2** (Equivalence & Consistency between optima of (1) and saddle point). *Suppose $\mathcal{X} = \mathbb{R}^n, \mathcal{Y} = \mathbb{R}^m$; $\rho_a, \rho_b$ are Borel probability distributions on $\mathcal{X}, \mathcal{Y}$; $\rho_a$ is atomless[4]. Assume the OT distance $C(\rho_a, \rho_b) < +\infty$ and the cost $c$ is continuous and bounded from below by a finite number $\underline{c}$, i.e. $c \geq \underline{c}$. Then,*

*(Equivalence) The following two statements are equivalent:*

1. *There exists at least one saddle point of $\mathcal{L}(T, f)$.*

2. *Both the primal problem* (1) *and the dual problem* (20) *admit at least one solution.*

*(Consistency) For any optimal solution $T_*$ to* (1), *optimal solution $\phi_*$ to* (20), *and saddle point $(\hat{T}, \hat{f})$ of $\mathcal{L}(T, f)$, we have $(T_*, \phi_*)$ is a saddle point of $\mathcal{L}(T, f)$, $\hat{T}$ is an optimal solution to* (1), *and $\hat{f}$ is an optimal solution to* (20). *Furthermore, $\mathcal{L}(\hat{T}, \hat{f}) = C_{Monge}(\rho_a, \rho_b)$.*

A direct corollary of Theorem 2 guarantees the existence of the saddle point.

**Corollary 1** (Existence of saddle point). *Suppose $\mathcal{X} = \mathbb{R}^n, \mathcal{Y} = \mathbb{R}^m$. Assume that $\rho_a, \rho_b$ are compactly supported Borel probability distributions on $\mathcal{X}, \mathcal{Y}$, and $\rho_a$ is absolute continuous with respect to the Lebesgue measure on $\mathbb{R}^n$. Assume $c$ satisfies* (16), (17), (18), (23), *and there exists a finite constant $\underline{c}$ such that $c \geq \underline{c}$. Then the Monge map $T_*$ and the optimal solution $\phi_*$ to* (20) *both exist and $(T_*, \phi_*)$ is a saddle point of $\mathcal{L}(T, f)$.*

---

[2]This means if $T_{**}$ is another Monge map solving the problem (1), then $T_* = T_{**}$ on $\mathrm{Spt}(\rho_a) \setminus E_0$, where $\mathrm{Spt}(\rho_a)$ is the support of $\rho_a$ and $E_0$ is a zero measure set.

[3]This is the joint distribution on $\mathcal{X} \times \mathcal{Y}$ obtained by pushing forward $\rho_a$ via the map $(\mathrm{Id}, T_*) : \mathcal{X} \to \mathcal{X} \times \mathcal{Y}$.

[4]This indicates that no single point in $\mathcal{X}$ carries a positive mass of $\rho_a$.

The existence of the saddle point of $\mathcal{L}(T, f)$ is a rather strong assumption and may not always hold (Zhang et al., 2022). We next provide some further assertions only focusing on the existence of sup-inf solution to (4) instead of the existence of saddle point.

**Theorem 3** (Consistency of sup-inf solution)**.** *Assume the cost function is continuous and satisfies the condition* (15)*. Suppose the sup-inf problem* (4) *admits at least one solution* $(\bar{T}, \bar{f})$*. Under the condition* $\bar{T}_\sharp \rho_a = \rho_b$*,* $\bar{f}$ *is an optimal solution to Kantorovich dual* (20)*, then* $\bar{T}$ *is optimal solution to Monge problem* (1)*; and furthermore,* $\mathcal{L}(\bar{T}, \bar{f}) = C_{Monge}(\rho_a, \rho_b)$*.*

It is hard to theoretically guarantee the condition $\bar{T}_\sharp \rho_a = \rho_b$ for general cost. Empirically, we verify our method can achieve $\bar{T}_\sharp \rho_a \approx \rho_b$ qualitatively (Figure 4, 5(b), 6, 7) and quantitatively (Figure 4(b), Table 1, 2). More results on consistency without $\bar{T}_\sharp \rho_a = \rho_b$ are presented in Appendix B.3.

In the implementation, we parametrize both the map $T$ and the dual variable $f$ by the neural networks $T_\theta, f_\eta$, with $\theta, \eta$ being the parameters. Consequently, our goal is to solve the following saddle point problem

$$\max_\eta \min_\theta \ \mathcal{L}(T_\theta, f_\eta) := \frac{1}{N} \sum_{k=1}^{N} c(X_k, T_\theta(X_k)) - f_\eta(T_\theta(X_k)) + f_\eta(Y_k) \qquad (6)$$

where $N$ is the size of the datasets and $\{X_k\}, \{Y_k\}$ are samples drawn by $\rho_a$ and $\rho_b$ separately. The algorithm is summarized in Algorithm 1. The computational complexity of our algorithm is similar with GAN-type methods. In particular, Algorithm 1 requires $\mathcal{O}(K(K_1 + K_2)B)$ operations in total.

---

**Algorithm 1** Computing optimal Monge map from $\rho_a$ to $\rho_b$

---

1: **Input**: Marginal distributions $\rho_a$ and $\rho_b$, Batch size $B$, Cost function $c(x, y)$.
2: Initialize $T_\theta, f_\eta$.
3: **for** $K$ steps **do**
4:     Sample $\{X_k\}_{k=1}^{B} \sim \rho_a$. Sample $\{Y_k\}_{k=1}^{B} \sim \rho_b$.
5:     Update $\theta$ to decrease (6) for $K_1$ steps.
6:     Update $\eta$ to increase (6) for $K_2$ steps.
7: **end for**
8: **Output**: The transport map $T_\theta$.

---

**Mapping between unequal dimensions** Our method is still applicable when $c(x, y)$ is defined on $\mathbb{R}^n \times \mathbb{R}^m$ and $n \neq m$. A typical example is $c(x, y) = \bar{c}(Q(x), y)$ (Pass, 2010; McCann & Pass, 2020), where $\bar{c}$ is defined on $\mathbb{R}^m \times \mathbb{R}^m$ and $Q(x)$ is an embedding function. To name a few, $Q(x)$ can be the upsampling function (Rout et al., 2022), or padding zero function (see Figure 8) as $n < m$, and a linear projection (Section 6.1) as $n > m$.

**Comparison with GAN** It is worth pointing out that our method and the WGAN (Arjovsky et al., 2017) are similar in the sense that they are both carrying out minimization over the generator/transport map and maximization over the discriminator/dual potential. However, there are two main distinctions, which are summarized in the following. More detailed discussions are postponed to Appendix C.

- (**Purpose**: arbitrary map vs optimal map) The purpose of WGAN is to find an *arbitrary map $T$* such that $T_\sharp \rho_a$ is close to the target distribution $\rho_b$; On the other hand, the purpose of our method is two-fold: we not only compute for the map $T$ that pushforwards $\rho_a$ to $\rho_b$, but also guarantee the *optimality* of $T$ in the sense of minimizing the total transportation cost $\mathbb{E}_{X \sim \rho_a} c(X, T(X))$.
- (**Designing logic**: minimizing distance vs computing distance itself) In WGAN, one aims at minimizing the OT distance as the discrepancy between $T_\sharp \rho_a$ and the target distribution $\rho_b$, the optimal value of the corresponding loss function thus equals to 0; In contrast, our proposed method computes OT distance (as well as the transport map) between source distribution $\rho_a$ and the target $\rho_b$. In this case, the optimal value of the associated loss function equals $C_{\text{Monge}}(\rho_a, \rho_b)$.

# 4 Error Estimation via Duality Gaps

In this section, we assume $\mathcal{X} = \mathcal{Y} = \mathbb{R}^d$, i.e. we consider Monge problem on $\mathbb{R}^d$. Suppose we solve (4) to a certain stage and obtain the pair $(T, f)$, inspired by Hütter & Rigollet (2020) and Makkuva et al. (2020), we provide an *a posteriori* estimate to a weighted $L^2$ error between our computed map $T$ and the optimal Monge map $T_*$. Before we present our result, we need the following assumptions.

**Assumption 1** (on cost $c(\cdot, \cdot)$). *We assume $c \in C^2(\mathbb{R}^d \times \mathbb{R}^d)$ is bounded from below. Furthermore, for any $x, y \in \mathbb{R}^d$, we assume $\nabla_x c(x, y)$ is injective w.r.t. $y$; $\nabla^2_{xy} c(x, y)$, as a $d \times d$ matrix, is invertible; and $\nabla^2_{yy} c(x, y)$ is independent of $x$.*

**Assumption 2** (on marginals $\rho_a$, $\rho_b$). *$\rho_a, \rho_b$ are compactly supported on $\mathbb{R}^d$, and $\rho_a$ has density.*

**Assumption 3** (on dual variable $f$). *Assume the dual variable $f \in C^2(\mathbb{R}^d)$ is always taken from c-concave functions, i.e., there exists certain $\varphi \in C^2(\mathbb{R}^d)$ such that $f(\cdot) = \inf_x \{\varphi(x) + c(x, \cdot)\}$ (see Definition 5.7 of Villani (2008)). Furthermore, we assume that there exists at least one minimizer $x_y \in \arg\min_x \{\varphi(x) + c(x, y)\}$ for any $y \in \mathbb{R}^d$. And the Hessian of $\varphi(\cdot) + c(\cdot, y)$ at $x_y$ is positive definite.*

**Assumption 4** (Infimum over $\tilde{T}$ achieves its minimizer). *Assume that there exists measurable $T_f \in \mathcal{M}(\mathbb{R}^d, \mathbb{R}^d)$ such that $T_f \in \arg\min_{\tilde{T}} \{\mathcal{L}(\tilde{T}, f)\}$. i.e., $T_f(x) \in \arg\min_y \{f(y) - c(x, y)\}$ for almost every $y \in \mathbb{R}^d$.*

For the sake of conciseness, let us introduce two notations. We denote $\sigma(x, y) = \sigma_{\min}(\nabla^2_{xy} c(x, y)) > 0$ as the minimum singular value of $\nabla^2_{xy} c(x, y)$; and denote

$$\lambda(y) = \lambda_{\max}(\nabla^2_{xx}(\varphi(x) + c(x, y))|_{x=x_y}) > 0 \tag{7}$$

as the maximum eigenvalue of the Hessian of $\varphi(\cdot) + c(\cdot, y)$ at $x_y$. We denote the duality gaps as

$$\mathcal{E}_1(T, f) = \mathcal{L}(T, f) - \inf_{\widetilde{T}} \mathcal{L}(\widetilde{T}, f), \quad \mathcal{E}_2(f) = \sup_{\widetilde{f}} \inf_{\widetilde{T}} \mathcal{L}(\widetilde{T}, \widetilde{f}) - \inf_{\widetilde{T}} \mathcal{L}(\widetilde{T}, f).$$

We can verify that the conditions mentioned in Theorem 1 are satisfied under Assumption 1 and Assumption 2. Thus the Monge map $T_*$ to the Monge problem (1) exists and is unique. We now state the main theorem on error estimation:

**Theorem 4** (*A posteriori* Error Estimation via Duality Gaps). *Suppose Assumption 1, 2, 3 and 4 hold. Let us further assume the sup-inf problem (4) admits a solution $(\bar{f}, \bar{T})$ that is consistent with the Monge problem, i.e. $\bar{T}$ equals $T_*$, $\rho_a$ almost surely. Then there exists a strict positive weight function $\beta(x) > \min_{y \in \mathbb{R}^m} \left\{ \frac{\sigma(x, y)^2}{2\lambda(y)} \right\}$ such that the weighted $L^2$ error between computed map $T$ and the Monge map $T_*$ is upper bounded by*

$$\|T - T_*\|_{L^2(\beta \rho_a)} \leq \sqrt{2(\mathcal{E}_1(T, f) + \mathcal{E}_2(f))}. \tag{8}$$

*The exact formulation of $\beta$ is provided in* (43) *in the appendix B.4.*

**Remark 1.** *The cost $c(x, y) = \frac{1}{2}\|x - y\|^2$ or $c(x, y) = -x \cdot y$ satisfy the conditions mentioned above. More specifically, when $c(x, y) = -x \cdot y$, we have $\sigma(x, y) = 1$, and $\lambda(y) = \frac{1}{\lambda_{\min}(\nabla^2 f(y))}$, which recovers Theorem 3.6 in Makkuva et al. (2020). Indeed, in this case, our $f, \lambda$ correspond to $f, \frac{1}{\alpha}$ in Makkuva et al. (2020).*

Theorem 4 provides an *a posteriori* error bound, i.e., for any computed $(T, f)$, we provide an upper bound (8) on the accuracy of $T$. The weight function $\beta(\cdot)$ relies on $f$, and $(T, f)$ could be arbitrary computed pair, thus it is almost impossible to provide a uniform lower bound for $\beta(\cdot)$, which is also dicussed in Makkuva et al. (2020, Remark 3.7). Further analysis on $\lambda(\cdot)$ and $\beta(\cdot)$ can be found in Appendix B.5. We clarify some limitations of our Theorem 4 below.

- (Hardness on evaluating $\mathcal{E}_1, \mathcal{E}_2$) It could be intractable to calculate $\mathcal{E}_2$ exactly due to the term $\sup_{\widetilde{f}} \inf_{\widetilde{T}} \mathcal{L}(\widetilde{T}, \widetilde{f})$, but one can approximate them by empirical values using samples after convergence.

- (Restricted form for analytic cost) Suppose that the cost $c$ satisfies Assumption 1 and is also analytic, then $c$ is restricted to the specific form $A(x) + F(x)^\top y + B(y)$, where $A, B$ are analytical functions on $\mathbb{R}^d$, and $F: \mathbb{R}^d \to \mathbb{R}^d$ is analytic with invertible Jacobian $DF(x)$ at any $x \in \mathbb{R}^d$.

Some further discussion on enforcing the c-concavity of the dual variable $f$ is presented in Appendix B.4.

# 5 Related work

Discrete OT methods (Courty et al., 2016; Feydy et al., 2020; Pooladian & Niles-Weed, 2021; Meng et al., 2019) solve the Kantorovich formulation with EMD (Nash, 2000) or Sinkhorn algorithm (Cuturi, 2013). It normally computes an optimal coupling of two empirical distributions. Such type of treatment has been widely accepted since it is friendly to high dimensional cases (Altschuler et al., 2017; Genevay et al., 2018; Li et al., 2019; Xie et al., 2020), but the algorithm does not scale well to a large number of samples and is not suitable to handle continuous probability measures. In addition, they cannot map the unseen samples. With an additional parameterization of a mapping function, such as linear or kernel function space (Perrot et al., 2016), they can map the unseen data. However, this map is not the solution to Monge problem but only an approximation of the optimal coupling (Courty et al., 2017, Sec. 2.1). Perrot et al. (2016) requires solving the matrix inverse when updating the transformation map, thus introducing a risk of instability. Their function space of the map is also less expressive than neural networks, which makes it difficult to solve the OT problem with complex transport costs and high dimension data. And they still can not handle large scale datasets.

With the rise of neural networks, neural OT has become popular with an advantage of dealing with continuous measure. Seguy et al. (2017); Genevay et al. (2016) solve the regularized OT problem, and as such introduce the bias. Another branch of work (Makkuva et al., 2020; Korotin et al., 2021b; Fan et al., 2020; Korotin et al., 2021c) comes with parameterizing Brenier potential by Input Convex Neural Network (ICNN) (Amos et al., 2017). The notable work (Makkuva et al., 2020) is special case of our formula. In particular, their dual formula reads $\sup_h \inf_g - \int [\langle x, \nabla g(x) - h(\nabla g(x)) \rangle] \rho_a dx - \int h(y) \rho_b dy$, where $g, h$ are ICNN. After a change of variable $\nabla g = T, h = -f + \| \cdot \|^2/2$, it becomes the same as our (5) equipped with the cost $c(x, y) = \|x - y\|^2/2$. Later, the ICNN parameterization was shown to be less suitable (Korotin et al., 2021a; Fan et al., 2022b; Korotin et al., 2022a) for large scale problems because ICNN is not expressive enough.

Recently, several works have illustrated the scalability of our dual formula with different realizations of the transportation costs. When $c(x, y) = \|x - y\|_2^2/2$, our method directly boils down to reversed maximum solver (Nhan Dam et al., 2019), which appears to be the best neural OT solvers among multiple baselines (Korotin et al., 2021a). Another variant (Rout et al., 2022) of our work obtains comparable performance in image generative models, which asserts the efficacy in unequal dimension tasks. Gazdieva et al. (2022) utilizes the same dual formula with more diverse costs in image super-resolution task.

Based on the similar dual formula, several works propose to add random noise as an additional input to make the map stochastic, i.e. one-to-many mapping, so it can approximate the Kantorovich OT plan. To make the learning of stochastic map valid, Korotin et al. (2022b) extends the transport cost $c(x, y)$ to be *weak cost* that depends on the mapped distribution. Typically, they involve the variance in the weak cost to enforce diversity. The stochastic map is, however, not suitable for our problem (1) because of the conditional collapse behavior (Korotin et al., 2022b, Sec 5.1). Later, Asadulaev et al. (2022) extends the dual formula to more general cost functional. We note that the term *general cost* in Asadulaev et al. (2022) is different from *general cost* in our paper. They propose to solve the General OT problem $\inf_{\pi \in \Pi(\rho_a, \rho_b)} \mathcal{F}(\pi)$ through the dual formula

$$\sup_f \inf_{\pi \in \Pi(\rho_a)} \{ \mathcal{F}(\pi) - \int_{\mathcal{Y}} f(y) d\pi(y) + \int_{\mathcal{Y}} f(y) d\rho_b(y) \}.$$

If $\mathcal{F}(\pi) = \int c(x, y) d\pi(x, y)$ and assuming optimal $\pi_*$ is of the form $\pi_* = (\mathrm{Id}, T_*)_\sharp \rho_a$, then their formula boils down to our (5). However, the assumption $\pi_* = (\mathrm{Id}, T_*)_\sharp \rho_a$ may not always hold.

# 6 Experiments

In this section, we specialize (5) with different general costs $c(x, y)$ to fit in various applications. We will not focus on the quadratic cost $c(x, y) = \|x - y\|_2^2$ since our formula in this special case has been extensively studied in multiple scenarios, e.g. generative model (Rout et al., 2022), super-resolution (Gazdieva et al., 2022), and style transfer (Korotin et al., 2022b). Instead, we focus on data with specific structure, which will be revealed in the cost function. Due to a wide span of tasks, it is unrealistic to find universal baselines in this section. We will select the proper baselines according to the scale of the examples.

## 6.1 Unpaired text to image generation

We consider the task of generating images given text prompts. Existing successful text-to-image generation algorithms (Ramesh et al., 2021; 2022; Saharia et al., 2022) are supervised learning methods, and as such rely on paired data. In real world, it can be exhausting to maintain large-scale paired datasets given that they are mostly web crawling data, and the validity period of web data is limited. We intend to learn a map between the text and image embedding space of the CLIP model without paired data. Our framework is shown in Figure 3. We use the same unpaired data generation scheme as Rout et al. (2022).

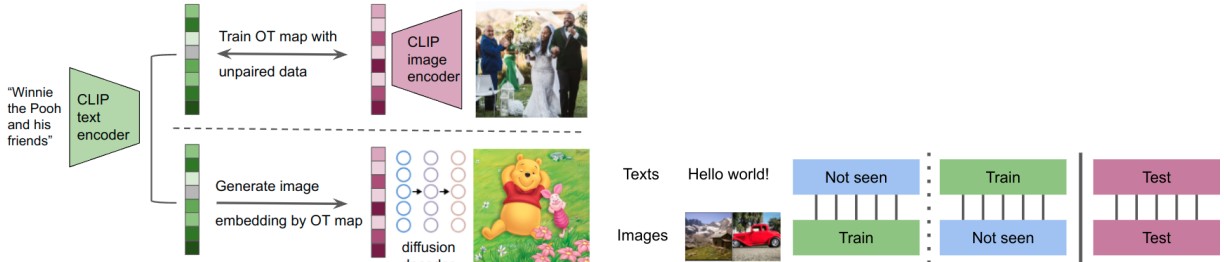

(a) Pipeline motivated by DALL·E2.        (b) Unpaired data generation process

Figure 3: (a) During the training process (above the dotted line), our map learns to generate image embeddings that maximize the expected similarity with input texts. During the evaluation (below the dotted line), given a text encoding from the CLIP model, our map outputs an image embedding, which conditions a pre-trained diffusion decoder to generate an image. The CLIP encoder and diffusion decoder are both pre-trained and frozen.
(b) We split a paired text-image dataset training into two parts: $\mathcal{S}$ and $\mathcal{T}$, and take the texts from $\mathcal{S}$ and images from $\mathcal{T}$ as our training data. Finally, we use the paired test data to compare with the real images.

In our algorithm, the sample from source distribution $\rho_a$ is the text encoding $x \in \mathbb{R}^{77 \times 768}$ ($x \neq 0$) from CLIP (Radford et al., 2021) model, and the sample from target distribution $\rho_b$ is the image embedding $y \in \mathbb{R}^{768}$, which is normalized to have $\|y\|_2 = 1$. The CLIP model is pre-trained on 400M (text, image) pairs of web data with contrastive loss, such that the paired texts and images share a large cosine similarity, and the non-paired ones present a small similarity. As a result, we choose the transport cost as the negative cosine similarity between $Rx$ and $y$:

$$c(x, y) = -\frac{\langle Rx, y \rangle}{\|Rx\|_2 \|y\|_2} = -\frac{\langle Rx, y \rangle}{\|Rx\|_2}. \tag{9}$$

The frozen matrix $R : \mathbb{R}^{77 \times 768} \rightarrow \mathbb{R}^{768}$ is extracted from a linear layer of CLIP model and it projects the text encoding $x$ to the same dimension as image embedding $y$. The normalized projected vector $Rx/\|Rx\|$ is called text embedding. Similar to DALL·E2 (Ramesh et al., 2021; 2022), we use text encoding instead of text embedding as the input data because the encoding contains more information. The cost $c(x, y)$ would enforce our map to generate an image embedding relevant to the input text.

We evaluate our algorithm on two datasets: Laion art and Conceptual captions 3M (CC-3M). Laion art dataset is filtered from a 5 billion dataset to have the high aesthetic level, which is suitable for the learning of image generation, while CC-3M is not curated. We use the OpenAI's CLIP (ViT-L/14) model and a DALL·E2 diffusion decoder. The training of their diffusion decoder requires paired data and its train dataset includes Laion art but not CC-3M. Therefore, our results on CC-3M are heavily based on the unpaired data because no part in Figure 3 has seen paired data of CC-3M, while on Laion art, the diffusion decoder has some paired data information gained from pre-training.

We show qualitative samples in Figure 1 and 2, where we compare with DALL·E2-Laion, a DALL·E2 model pre-trained on the paired Laion aesthetic dataset, which includes Laion art dataset. Figure 1 confirms that our transport map can generate image embeddings with comparable quality even without paired data. Since CC-3M is a zero-shot dataset for DALL·E2-Laion, the performance of DALL·E2-Laion clearly drops in

Figure 2, while our model still generates reasonable images. We visually verify our map achieves $T_\sharp \rho_a \approx \rho_b$ by comparing the generated embeddings with the real image embeddings (on a test dataset). In Figure 4 (a), we present the results of our method, along with the results of a non-linear kernel map given by Perrot et al. (2016). While the latter method struggles to generalize to unseen data, our method is able to generate a distribution that closely matches the ground truth.

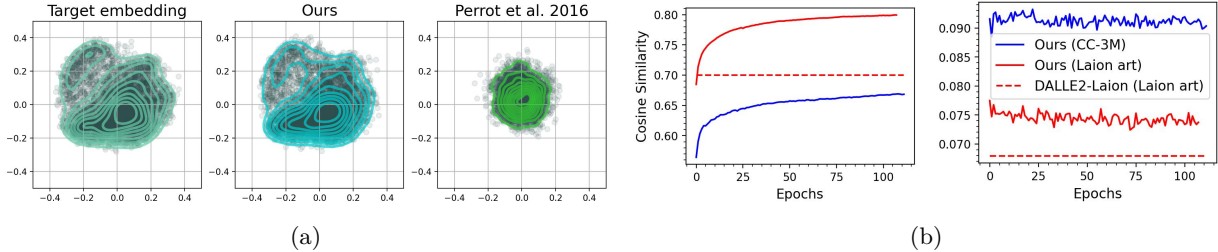

(a)     (b)

Figure 4: (a) Comparison between the Laion art target image embeddings and fitted measure of generated embeddings by our method and Perrot et al. (2016). Samples are projected onto 2D plane by PCA.
(b) Averaged cosine similarity between the generated image embeddings and (Left) the ground truth image embeddings or (Right) the unrelated text embeddings. The similarity on the left depicts how well the model recovers the real image embedding, and the similarity on the right quantifies the overfitting behavior (the lower the better).

We also quantitatively compare with the baseline in terms of cosine similarity in Figure 4 (b). Note that we also normalize $T(x)$, so $T(x) = y \Leftrightarrow$ cos-similarity$(T(x), y) = 1$. Our method achieves higher cosine similarity w.r.t. the real images on Laion art dataset. In practice, we observe that the relevance of each (text, image) pair in CC-3M is more noisy than Laion art, which makes the model more difficult to learn the real image embedding. This explains why our cosine similarity w.r.t real image embedding on CC-3M can not converge to the same level as Laion art. In the meantime, our overfitting level is low on both datasets.

## 6.2 Class-preserving map

We consider the class-preserving map between two labelled datasets. The input distribution $\rho_a = \sum_{j=1}^{J} a_j \rho_a^j$ is a mixture of $J$ distinct distributions $\{\rho_a^j\}_{j=1}^{J}$. Similarly, the target distribution is $\rho_b = \sum_{j=1}^{J} b_j \rho_b^j$, where $\{a_j\}$ and $\{b_j\}$ satisfy $\sum_j a_j = 1$ and $\sum_j b_j = 1$. Each distribution $\rho_a^j$ (or $\rho_b^j$) is associated with a **known** label/class $j$. We further assume that the support of $\{\rho_a^j\}_j$ (or $\{\rho_b^j\}_j$) are disjoint with respect to $j$. We seek a map that solves the problem

$$\min_{T_\sharp \rho_a^j = \rho_b^j} \int_{\mathbb{R}^n} \|x - T(x)\|^2 \rho_a(x) \; dx, \quad \forall j \tag{10}$$

where the constraint asks the map to preserve the original class. To approximate this map, we replace the constraint by a contrastive penalty. This trick transfers (10) to the original Monge problem with a cost

$$c(\{x, y\}, \{x', y'\}) = \|x - x'\|^2 + \mathbf{1}(y \neq y'), \tag{11}$$

where $y$ and $y'$ are the labels corresponding to $x$ and $x'$, $\mathbf{1}$ is the indicator function. To better guide the mapping, we involve the label as an input to the mapping $T$ and the potential $f$. Denote $\ell(\cdot) : \mathbb{R}^m \to \mathbb{R}^J$ as a pre-trained classifier on the target domain $\rho_b$. Given a feature from the target domain, $\ell(\cdot)$ will output a probability vector. Then, our full formula reads

$$\sup_f \inf_T \int \left[ \|x - \bar{x}\|^2 - y^\top \bar{y} - f(\bar{x}; \bar{y}) \right] d\rho_a + \int f(x'; y') d\rho_b, \quad \text{where} \quad \bar{x} = T(x; y), \quad \bar{y} = \ell(T(x; y)).$$

In practice, $y \in \mathbb{R}^J$ is a one-hot label and it is known because both the source and target datasets are labeled.

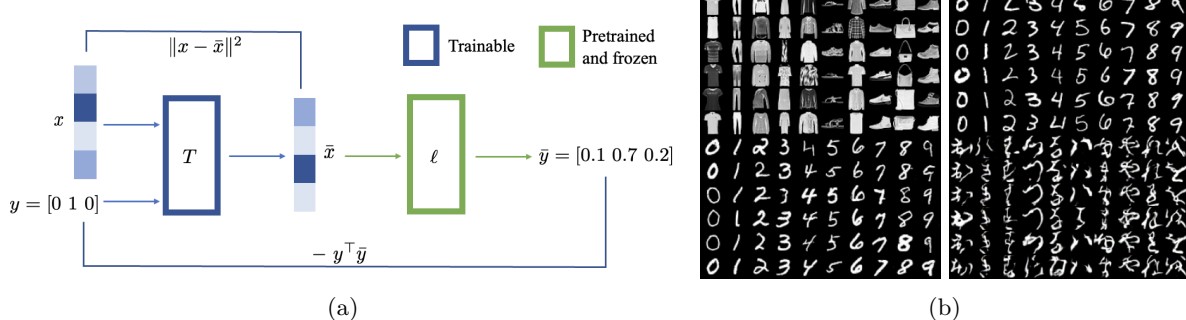

(a)                                                      (b)

Figure 5: (a) Our cost is composed of the ordinary cost $\|x - \bar{x}\|^2$ and the contrastive label cost $-y^\top \bar{y}$. (b) Class-preserving mapping by our conditional map. The top block is the source images, and the bottom block is the pushforward images. Each column represents a class. Left: F $\to$ M, Right: M $\to$ K.

Table 1: Class-preserving map result: accuracy of the maps and FID of generated samples. Our map is deterministic. The map by Asadulaev et al. (2022) is non-conditional on the label. M, K, F represent MNIST, KMNIST, FMNIST correspondingly. Our accuracy result is significantly better than Asadulaev et al. (2022) thanks to the guidance of contrastive cost (11).

|  |  | Asadulaev et al. (2022) deterministic map | Asadulaev et al. (2022) stochastic map | Ours with conditional map | Ours with non-conditional map |
|---|---|---|---|---|---|
| FID | M $\to$ K | 17.26 | **9.69** | 18.07 | 17.45 |
|  | F $\to$ M | 7.15 | **5.26** | 5.78 | 6.86 |
| Accuracy | M $\to$ K | 79.20 | 61.91 | **99.59** | **93.02** |
|  | F $\to$ M | 82.97 | 83.22 | **99.73** | **98.93** |

Recent two works (Asadulaev et al., 2022; Bunne et al., 2022) are along this line of class-guided OT maps. The former one learns an unconditional map $T(x)$, i.e. no need of label information during the inference, and the latter one learns a conditional map $T(x; y)$. Our method is different from both of them because their costs do not explicitly involve the label, while we include the label in the contrastive cost (11). Our method is designed to learn a conditional map, but we can also fit an unconditional map to the learned conditional map for the convenience of inference.

We compare our algorithm with Asadulaev et al. (2022) on $^*$NIST (LeCun & Cortes, 2005; Xiao et al., 2017; Clanuwat et al., 2018) datasets, where we use default labels in PyTorch. The visualization of our mapping is presented in Figure 5 (b). We also calculate FID (Heusel et al., 2017) between the mapped source test dataset and the target test dataset. To quantify how well the map preserves the original class, we use a pre-trained SpinalNet classifier (Kabir et al., 2022) on the target domain and evaluate the accuracy of the predicted label. If the predicted label of $T(x; y)$ equals to the original label $y$, then the mapping is correct, otherwise wrong. We show the quantitative result in Table 1, where the results of Asadulaev et al. (2022) are from their paper. Since Asadulaev et al. (2022) makes inference using an unconditional map, to make a fair comparison, we present the results with a fitted unconditional map as well in Table 1. The stochastic map in Asadulaev et al. (2022) leads to a lower FID because the loss function for the stochastic map incorporates the "variance" of the pushforward distribution, which encourages the generation of more diverse data. However, because the stochastic map requires random noise as input, it requires a more sophisticated network architecture and is more computationally expensive to train.

## 6.3 Unpaired image inpainting

In this section, we show the effectiveness of our method on the inpainting task with random rectangle masks. We take the distribution of occluded images to be $\rho_a$ and the distribution of the full images to be $\rho_b$. In many inpainting works, it's assumed that an unlimited amount of paired training data is accessible (Zeng

et al., 2021). However, some real-world applications do not involve the paired datasets. Accordingly, we consider the unpaired inpainting task. The training and test data are generated in the same way as Figure 3 (b). We choose cost function to be mean squared error (MSE) in the unmasked area

$$c(x, y) = \alpha \cdot \frac{\|x \odot M(x) - y \odot M(x)\|_2^2}{n},$$  (12)

where $M(x)$, depending on $x$, is a binary mask with the same size as the image. $M$ takes the value 1 in the unoccluded region, and 0 in the unknown/missing region. $\odot$ represents the point-wise multiplication, $\alpha$ is a tunable coefficient, and $n$ is dimension of $x$. Different from Rout et al. (2022), where $c(x, y) = \|x - y\|^2/n$, we include the mask in the cost to avoid the penalty in the masked region. Otherwise, it penalizes the generation of non-zero pixels because the pixel values are zeros in the masked region of source image. The map satisfying constraints $T_\sharp \rho_a = \rho_b$ and $T(x) \odot M(T(x)) = x \odot M(x)$ is the optimal map.

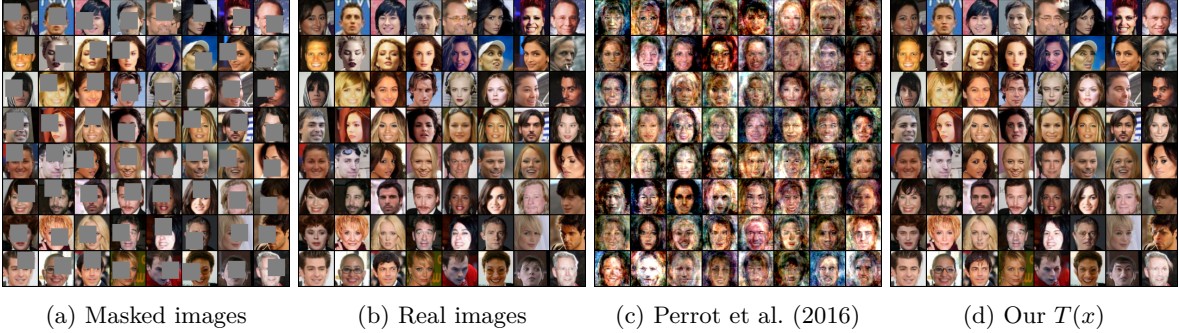

| (a) Masked images | (b) Real images | (c) Perrot et al. (2016) | (d) Our $T(x)$ |

Figure 6: Unpaired image inpainting on **test** dataset of CelebA $64 \times 64$.

We conduct the experiments on CelebA $64 \times 64$ and $128 \times 128$ datasets (Liu et al., 2015). The input images ($\rho_a$) are occluded by randomly positioned square masks. Each of the source $\rho_a$ and target $\rho_b$ distributions contains $80k$ images. We present the empirical results of inpainting in Figure 6 and 15. In Figure 6, we compare with the discrete OT method Perrot et al. (2016) since it also accepts general costs. It fits a map that approximates the barycentric projection so it can also provide out-of-sample mapping, although their map is not the solution to our Monge problem. We use 1000 unpaired samples for Perrot et al. (2016) to learn the optimal coupling. The comparison in Figure 6 confirms that our recovered images have much better quality than the discrete OT method.

We also evaluate FID of the generated composite images w.r.t. the original images on the test dataset. We use $40k$ images and compute the score with the implementation provided by Obukhov et al. (2020). We choose transport map with $L^2$ cost and WGAN-GP (Gulrajani et al., 2017) as the baselines and show the comparison results in Table 2. Our essential difference with transport map with $L^2$ cost is that the latter cost does not involve masks, which jeopardizes the performance when the position of masks are changeable.

Table 2: Quantitative evaluation results on CelebA $64 \times 64$ test dataset.

|  | Cost (12) | | Cost $\alpha\|x - y\|^2/n$ | WGAN-GP |
|---|---|---|---|---|
|  | $\alpha = 10$ | | $\alpha = 10^4$ | |
| FID | 9.2857 | **3.7109** | 8.9398 | 6.7479 |

Theoretically, the change of $\alpha$ should not affect the OT map because $f$ should be able to scale arbitrarily. However, in practice, due to the self-regularization of neural networks (Martin & Mahoney, 2021), $f$ can not scale arbitrarily with $\alpha$. As a result, modifying $\alpha$ will change the loss landscape. During the training, we observe that the magnitude ratio $r = |\mathbb{E}[c(x, T(x))]|/|\mathbb{E}[f(T(x))]|$ decreases from 18 to 0.2 when $\alpha$ drops from $10^4$ to 10. We also observe the map learned with a larger $\alpha$ can generate more realistic images with natural transition in the mask border and exhibit more details on the face (see Figure 14 in the appendix). In our experiments, we observe that $r > 10$ yields reasonable results.

### 6.4 Population transportation on the sphere

Motivated by the recent spherical transportation works (Cohen et al., 2021; Amos et al., 2022), we consider the following population transport problem on the sphere for testing our proposed method. It is well-known that the population on earth is not distributed uniformly over the land due to various factors such as landscape, climate, temperature, economy, etc. Suppose we ignore all these factors, and we would like to design an optimal transport plan under which the current population travels along the earth's surface to form a rather uniform (in spherical coordinate) distribution over the earth's landmass. We treat the earth as an ideal sphere with radius 1, then we are able to formulate such problem as a Monge problem defined on $D = [0, 2\pi) \times [0, \pi]$ with certain cost function $c$. To be more specific, we consider the spherical coordinate $(\theta, \phi)$ on $D$, which corresponds to the point $(\cos\theta\sin\phi, \sin\theta\sin\phi, \cos\phi)$ on the sphere. For any $(\theta_1, \phi_1), (\theta_2, \phi_2) \in D$, we set the distance $c(\cdot, \cdot)$ function as the *geodesic distance* on sphere, which is formulated as

$$c((\theta_1, \phi_1), (\theta_2, \phi_2)) = \arccos(\sin\phi_1\sin\phi_2\cos(\theta_1 - \theta_2) + \cos\phi_1\cos\phi_2). \tag{13}$$

Then assume the current population distribution over the sphere is denoted as $\rho_a$ (in Cartesian coordinate), which introduces the corresponding distribution $\rho_a^{\text{Sph}}$ (in sperical coordinate) on $D$. Denote $D_{\text{land}} \subset D$ as the region of the land in the spherical coordinate system. We set the target distribution $\rho_b^{\text{Sph}}$ as the uniform distribution supported on $D_{\text{land}}$. We aim at solving the following Monge problem

$$\min_{T: T_\sharp \rho_a^{\text{Sph}} = \rho_b^{\text{Sph}}} \left\{ \int_D c((\theta, \phi), T(\theta, \phi)) \rho_a^{\text{Sph}} \, d\theta d\phi \right\}.$$

The samples used in our experiment are generated from the licensed dataset from Doxsey-Whitfield et al. (2015). In our exact implementation, in order to avoid the explosion of the gradient of $\arccos(\cdot)$ near $\pm 1$, we replace the cost $c$ with its linearization (In our computation, the constant $\frac{\pi}{2}$ can be omitted.)

$$\widehat{c}((\theta_1, \phi_1), (\theta_2, \phi_2)) = \frac{\pi}{2} - (\sin\phi_1\sin\phi_2\cos(\theta_1 - \theta_2) + \cos\phi_1\cos\phi_2). \tag{14}$$

Furthermore, to guarantee that each sample point is transported onto landmass, we composite the trained map $T$ with a map $\tau : D \to D$ that remains samples on land unchanged but maps samples on sea back to the closest location on land among randomly selected sites. In Figure 7, we compare our result with the linear transformation method introduced in Perrot et al. (2016). For more general kernel map, such as Gaussian kernel, their algorithm is not very stable and it is very difficult to obtain valid results.

Although the ground truth for this example is unknown to us, we refer the reader to Appendix D for more synthetic examples with ground truth such as distributions with unequal dimensions, cost of decreasing functions, and Monge map on sphere.

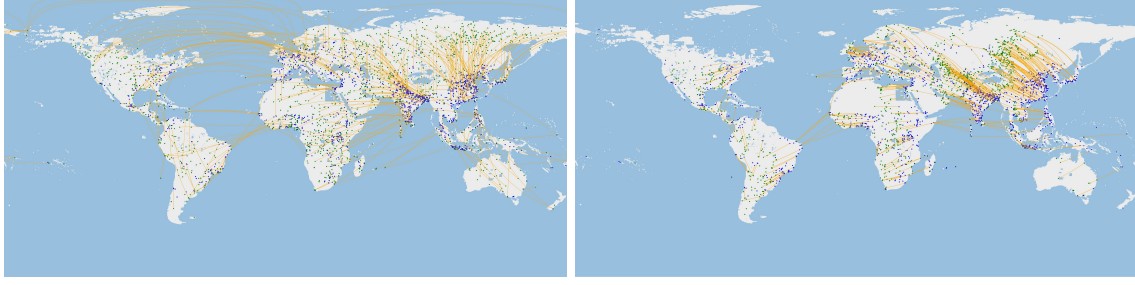

Figure 7: We show our result on the left and Perrot et al. (2016) on the right. Each figure plots samples from the source distribution $\rho_a^{\text{Sph}}$ (blue) and samples from the pushforward distribution $T_{\theta\sharp}\rho_a^{\text{Sph}}$ (green). We also demonstrate the computed transport map (orange) for the first 500 randomly generated points from source $\rho_a^{\text{Sph}}$ to target $\rho_b^{\text{Sph}}$.

# 7   Conclusion and Discussion

In this paper, we present a novel method to compute the Monge map between two given distributions with flexible transport cost functions. In particular, we consider applying Lagrange multipliers on the Monge problem, which leads to a sup-inf saddle point problem. By further introducing neural networks into our optimization, we obtain a scalable algorithm that can handle most general costs.

**Broader impact**   Our method will become a useful tool for machine learning applications such as domain adaption, and image restoration that requires transforming data distributions. Compared to other deep learning methods, our scheme reduces the need for paired data; compared to other OT methods, our scheme can be applied to high dimensional large-scale datasets, and we do not involve regularization which could hurt the quality of transformed data. In the future, it is promising to specialize our method to structured data, such as time series, graphs, point clouds, and SPD manifolds (Ju & Guan, 2022).

**Limitation**   Firstly, We observe that when the target distribution is a discrete distribution with fixed support, our method tends to be more unstable. We conjecture that this is because the network parameterization of the discriminator is too complex for this type of uncomplicated target distribution. Secondly, there could be a gap between the theory and some of our experiments. We make a summary table in Section F, which demonstrates that for some experiments, the existence and uniqueness of the Monge map are not guaranteed. It is either because the space is not regular or the assumptions in Theorem 1 are not fully satisfied. We leave the theoretical study for those examples as the future direction.

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

## Contents

# A    Backgrounds of optimal transport problem with general cost function

We present some fundamental results of OT problem with general cost in this section. Most of the results are collected and summarized from Villani (2008).

Suppose $M$ is a smooth, complete connected Riemmannian manifold. In most of our discussions, we treat $M$ as the Euclidean space $\mathbb{R}^n$ for the sake of convenience. Let us denote $\mathcal{X}$ as a closed subset of $M$. Thus $\mathcal{X}$ is a Polish space, meaning $\mathcal{X}$ is a separable[5], complete, metric space. We assume $\mathcal{Y}$ is a general Polish space. Suppose the cost function $c$ is defined on $\mathcal{X} \times \mathcal{Y}$. Suppose $\mathcal{X}, \mathcal{Y}$ are equipped with their Borel $\sigma$-algebra. Denote $\mathscr{P}(\mathcal{X}), \mathscr{P}(\mathcal{Y})$ as the space of Borel probability measures on $\mathcal{X}, \mathcal{Y}$. We mainly focus on the Monge problem (1) from $\rho_a \in \mathscr{P}(\mathcal{X})$ to $\rho_b \in \mathscr{P}(\mathcal{Y})$ with cost function $c$.

Before the detailed discussion, we list some useful conditions that may be used later.

A)  $c$ is bounded below, i.e., there exists $a \in L^1(\mathcal{X}), b \in L^1(\mathcal{Y})$, s.t. $c(x,y) \geq a(x) + b(y)$ on $\mathcal{X} \times \mathcal{Y}$;    (15)

B)  $c$ is *locally Lipschitz* as a function of $x$ on $\text{int}(\mathcal{X})$, locally in $y$;    (16)

C)  $c$ is everywhere *superdifferentiable* as a function of $x \in \text{int}(\mathcal{X})$, for all $y$;    (17)

D)  For arbitrary $x \in \text{int}(\mathcal{X}), \nabla_x c(x, \cdot) : \mathcal{Y} \to \mathbb{R}^n$,  is injective,    (18)

   i.e., suppose for $x, y, y', \nabla_x c(x,y) = \nabla_x c(x, y')$, then $y = y'$.

Here we denote $\text{int}(\mathcal{X})$ as the interior of the set $\mathcal{X}$, i.e., $\text{int}(\mathcal{X})$ is the union of all open sets in $M$ contained in $\mathcal{X}$. And we define the local Lipschitz property and superdifferentiablity as follows.

**Definition 1** (Locally Lipschitz). *Let $U \subset \mathbb{R}^n$ be open and let $f : \mathbb{R}^n \to \mathbb{R}$ be given. Then*

*(1) $f$ is Lipschitz if there exists $L < \infty$ such that*

$$\forall x, z \in \mathbb{R}^n, \quad |f(z) - f(x)| \leq L|x - z|.$$

*(2) $f$ is said to be locally Lipschitz if for any $x_0 \in \mathbb{R}^n$, there is a neighbourhood of $x_0$ in which $f$ is Lipschitz.*

**Definition 2** (Superdifferentiablity). *$f : \mathbb{R}^n \to \mathbb{R}$ is superdifferentiable at $x$, with supergradient $p$, if*

$$f(z) \leq f(x) + \langle p, z - x \rangle + o(|z - x|).$$

*Here the little-o notation indicates that $\lim_{z \to x} \frac{o(|z-x|)}{|z-x|} = 0$.*

The relation between optimal values of Monge problem (1) and its relaxation, namely, the general OT problem (2) is studied in Pratelli (2007) (c.f. Bibliographical notes of Chap 4 in Villani (2008)). We summarize the result below.

**Theorem 5.** *Suppose $\mathcal{X}, \mathcal{Y}$ are two Polish spaces; $\rho_a, \rho_b$ are Borel probability distributions on $\mathcal{X}, \mathcal{Y}$. Suppose $\rho_a$ is atomless (i.e. no single point carries a positive mass). Assume further $c$ is continuous and satisfies (15). Then $C_{Monge}(\rho_a, \rho_b) = C(\rho_a, \rho_b)$.*

Recall that the Kantorovich dual problem of the primal OT problem (2) is formulated as

$$K(\rho_a, \rho_b) := \sup_{\substack{(\psi, \phi) \in C_b(\mathcal{X}) \times C_b(\mathcal{Y}) \\ \phi(y) - \psi(x) \leq c(x,y) \; \forall \; x \in \mathcal{X}, y \in \mathcal{Y}}} \left\{ \int_{\mathcal{Y}} \phi(y) \rho_b(y) \, dy - \int_{\mathcal{X}} \psi(x) \rho_a(x) \, dx \right\}.$$

We denote the optimal value of this dual problem as $K(\rho_a, \rho_b)$.

It is not hard to tell that the above dual problem is equivalent to any of the following two problems

$$\sup_{\psi \in C_b(\mathcal{X})} \left\{ \int_{\mathcal{Y}} \psi^{c,+}(y) \rho_b(y) \, dy - \int_{\mathcal{X}} \psi(x) \rho_a(x) \, dx \right\}, \tag{19}$$

---

[5]A topological space is separable if it contains a countable dense set. Since $\mathbb{R}^n$ is separable, its subset $\mathcal{X}$ equipped with the subspace topology is also separable.

$$\sup_{\phi \in C_b(\mathcal{Y})} \left\{ \int_{\mathcal{Y}} \phi(y)\rho_b(y) \; dy - \int_{\mathcal{X}} \phi^{c,-}(x)\rho_a(x) \; dx \right\}. \tag{20}$$

Here the $c$-transform $\psi^{c,+}$, $\phi^{c,-}$ are defined via infimum/supremum convolution as:

$$\psi^{c,+}(y) = \inf_x (\psi(x) + c(x,y)); \tag{21}$$

$$\phi^{c,-}(x) = \sup_y (\phi(y) - c(x,y)). \tag{22}$$

If we denote $\psi_*, \phi_*$ as the optimal solutions to (19), (20) respectively. Then both $(\psi_*, \psi_*^{c,+})$ and $(\phi_*^{c,-}, \phi_*)$ are the optimal solutions to (3); $\psi_*^{c,+}$, $\phi_*^{c,-}$ are also solutions to (20), (19) respectively.

**Remark 2** (c.f. Theorem 5.10 of Villani (2008)). *It is worth mentioning that one can relax the $C_b(\mathcal{X}), C_b(\mathcal{Y})$ spaces in (3), (19) and (20) to $L^1(\mathcal{X}; \rho_a)$ and $L^1(\mathcal{Y}; \rho_b)$.*

The following theorem states the equivalent relationship between the primal OT problem (2) and its Kantorovich dual (3). The proof of a more general version can be found in Theorem 5.10 of Villani (2008).

**Theorem 6** (Kantorovich Duality). *Suppose $\mathcal{X}, \mathcal{Y}$ are two Polish spaces. Suppose $c$ is a cost function defined on $\mathcal{X} \times \mathcal{Y}$, $c$ is continuous and satisfies (15). Recall that $C(\rho_a, \rho_b)$ denotes the infimum value of (2) and $K(\rho_a, \rho_b)$ denotes the maximum value of (3) (or equivalently, (19), (20)). Then $C(\rho_a, \rho_b) = K(\rho_a, \rho_b)$. Furthermore, if we assume that $c$ is bounded from above by*

$$c(x,y) \le u(x) + v(y), \;\; with \; u \in L^1(\mathcal{X}; \rho_a), v \in L^1(\mathcal{Y}; \rho_b). \tag{23}$$

*Then the optimal transport plan $\pi_*$ for (2) and optimal potential $\phi_*$ for (20) both exist.*

The following result states the existence and uniqueness of the optimal solution to the Monge problem. It also reveals the relation between the optimal map $T_*$ of (1) and the optimal transport plan $\pi_*$ of (2). It is a simplified version of Theorem 10.28 combined with Theorem 10.26 taken from Villani (2008).

**Theorem 1** (Existence, uniqueness and characterization of the optimal Monge map). *Assume $\mathcal{X}, \mathcal{Y}, \rho_a, \rho_b$ are defined as in the beginning of section 2. Assume the dimension of boundary $\partial \mathcal{X}$ is no larger than $n-1$. Suppose the cost $c$ is continuous and satisfies (15), (16), (17), (18), we further assume that $\rho_a$ and $\rho_b$ are compactly supported and $\rho_a$ is absolutely continuous with respect to the Lebesgue measure on $\mathbb{R}^n$. Then there exists a unique-in-law transport map $T_*$ solving the Monge problem (1). And the joint distribution $(Id, T_*)_\sharp \rho_a$ on $\mathbb{R}^{n \times m}$ is the optimal transport plan of the general OT problem (2). Furthermore, there exists a $c$-convex[6] $\psi_*$, which can be proved to be differentiable $\rho_a$ almost surely[7], and $\nabla \psi_*(x) + \nabla_x c(x, T_*(x)) = 0$, $\rho_a$ almost surely.*

**Remark 3.** *The differentiable property of $c$-convex $\psi_*$ is a direct result of Theorem 10.26 from Villani (2008).*

**Remark 4.** *The condition $C(\rho_a, \rho_b) < +\infty$ in Theorem 10.28 of Villani (2008) is automatically satisfied since we assume that $\rho_a, \rho_b$ are compactly supported and the cost $c$ is continuous. It is also worth mentioning that the conditions in Theorem 1 can be modified and we can still obtain the existence and uniqueness of the Monge map. We refer the readers to Remark 10.33 of Villani (2008) for more details.*

**Remark 5** (Discussion on regularity of $T_*, \psi_*$). *From the above theorem, one can tell that the regularity of the Monge map $T_*$ is closely related to the regularity of $\psi_*$, which is a very important topic in OT and the Monge-Ampère equation associated to the OT problem. We refer the reader to Theorem 12.50, 12.51 and 12.52 of Villani (2008) for details. Furthermore, in recent research such as Hütter & Rigollet (2020), Manole et al. (2022), stronger $C^{k,\alpha}$ regularity on $\psi_*$ were used to establish bounds on minimax rates for certain types of estimators under the statistical setting. In our paper, we did not consider the aforementioned statistical problem, so we omit the detailed discussion on the regularities of both $T_*$ and $\psi_*$.*

---

[6]C.f. the definition in Assumption 3.

[7]$\psi_*$ is differentiable at $x$ for all $x \in \mathrm{Spt}(\rho_a) \setminus E_0$, where $E_0$ is a zero measure set.

# B    Proof of theoretical results

We first define the saddle point of $\mathcal{L}(T, f)$ to the sup-inf problem (4) as following.

**Definition 3** (Saddle point). *Suppose $\mathcal{X}, \mathcal{Y}$ are two Polish spaces. We say $(\hat{T}, \hat{f})$ is a saddle point of the functional $\mathcal{L}(T, f)$ to the sup-inf problem (4) defined in (5) if $(\hat{T}, \hat{f})$ satisfy*

$$\hat{T} \in \operatorname*{argmin}_{T \in \mathcal{M}(\mathcal{X}, \mathcal{Y})} \{\mathcal{L}(T, \hat{f})\}; \quad \hat{f} \in \operatorname*{argmax}_{f \in C_b(\mathcal{Y})} \{\mathcal{L}(\hat{T}, f)\}.$$

Before we introduce the main results, we present a useful lemma that enables one to switch the optimization over function parameters and the integral. Such result is a corollary of Theorem 14.60 in Rockafellar & Wets (2009) (also c.f. Theorem 3A in Rockafellar (1976)) and it is also used in (Korotin et al., 2022b; Gazdieva et al., 2022).

**Lemma 1.** *Let $f : \mathbb{R}^n \times \mathbb{R}^m \to \mathbb{R}$ is continuous function. Let $\mu$ be a Borel measure defined on $\mathbb{R}^n$. Denote $\mathcal{M}(\mathbb{R}^n, \mathbb{R}^m)$ as the space of all measurable functions $T : \mathbb{R}^n \to \mathbb{R}^m$. Then we have*

$$\inf_{T \in \mathcal{M}(\mathbb{R}^n, \mathbb{R}^m)} \int_{\mathbb{R}^n} f(x, T(x)) d\mu(x) = \int_{\mathbb{R}^n} \inf_{\xi \in \mathbb{R}^m} f(x, \xi) \ d\mu(x).$$

## B.1    Proof of Theorem 2

*Proof.* $(1 \to 2)$ First denote $T_*$ as a solution to (1) and $\phi_*$ as a solution to (20). We prove $(T_*, \phi_*)$ is the saddle point of $\mathcal{L}(T, f)$. We first prove $T_* \in \operatorname*{argmin}_{T \in \mathcal{M}(\mathcal{X}, \mathcal{Y})} \{\mathcal{L}(T, \phi_*)\}$. By using lemma 1, one gets

$$\inf_T \{\mathcal{L}(T, \phi_*)\} = -\int_{\mathcal{X}} \sup_{\xi} \{\phi_*(\xi) - c(x, \xi)\} \rho_a(x) dx + \int_{\mathcal{Y}} \phi_*(y) \rho_b(y) dy = \int_{\mathcal{Y}} \phi_*(y) \rho_b(y) dy - \int_{\mathcal{X}} \phi_*^{c,-}(x) \rho_a(x) dx.$$

Since $\phi_*$ solves (20), the right-hand side of the above equals $K(\rho_a, \rho_b)$. Now by Theorem 5, 6, one can obtain $C_{\text{Monge}}(\rho_a, \rho_b) = C(\rho_a, \rho_b) = K(\rho_a, \rho_b)$, which leads to the strong duality of such Monge problem. Thus we have $\inf_T \{\mathcal{L}(T, \phi_*)\} = C_{\text{Monge}}(\rho_a, \rho_b)$. Since $T_*$ solves (1), we have

$$\inf_T \{\mathcal{L}(T, \phi_*)\} = \int_{\mathcal{X}} c(x, T_*(x)) \rho_a(x) dx = \mathcal{L}(T_*, \phi_*).$$

The second equality above is due to the constraint $T_{*\sharp} \rho_a = \rho_b$. This proves $T_* \in \operatorname*{argmin}_{T \in \mathcal{M}(\mathcal{X}, \mathcal{Y})} \{\mathcal{L}(T, \phi_*)\}$.

The proof of $\phi_* \in \operatorname*{argmax}_{\phi \in C_b(\mathcal{Y})} \{\mathcal{L}(T_*, \phi)\}$ is straightforward, since $\mathcal{L}(T_*, \phi) = \int_{\mathcal{X}} c(x, T_*(x)) \rho_a(x) dx$ is independent of $\phi$.

Thus we have proved that $(T_*, \phi_*)$ is a saddle of $\mathcal{L}(T, f)$.

$(2 \to 1)$ Let us first prove that $\hat{T}$ is an optimal solution to Monge problem (1). We first claim that $\hat{T}_{\sharp} \rho_a = \rho_b$. Suppose this is not true, then one can find a function $\varphi \in C_b(\mathcal{Y})$ such that $\int_{\mathcal{Y}} \varphi(\hat{T}(x)) \rho_a(x) dx \neq \int_{\mathcal{Y}} \varphi(y) \rho_b(y) dy$[8]. Without loss of generality, assume $\int_{\mathcal{Y}} \varphi(y) \rho_b(y) dy - \int_{\mathcal{X}} \varphi(\hat{T}(x)) \rho_a(x) dx > 0$. We pick the test function $\nu \varphi \in C_b(\mathcal{Y})$ with $\nu \in \mathbb{R}$ as an arbitrary real number, then we consider

$$\mathcal{L}(\hat{T}, \nu \varphi) = \int_{\mathcal{X}} c(x, \hat{T}(x)) \rho_a(x) dx + \nu (\underbrace{\int_{\mathcal{Y}} \varphi(y) \rho_b(y) dy - \int_{\mathcal{X}} \varphi(\hat{T}(x)) \rho_a(x) dx}_{\text{denote as } \delta > 0}) \geq \underline{c} + \mu \delta. \tag{24}$$

Recall $\hat{T} \in \operatorname*{argmin}_T \{\mathcal{L}(T, \hat{f})\}$, use Lemma 1, then we have

$$\mathcal{L}(\hat{T}, \hat{f}) = \inf_T \{\mathcal{L}(T, \hat{f})\} = \int_{\mathcal{Y}} \hat{f}(y) \rho_b(y) dy - \int_{\mathcal{X}} \hat{f}^{c,-}(x) \rho_a(x) dx \leq K(\rho_a, \rho_b) = C(\rho_a, \rho_b) < +\infty. \tag{25}$$

---

[8]This is due to the fact that when $\mathcal{Y}$ is a metric space, and $\mu_a, \mu_b \in \mathscr{P}(\mathcal{Y})$ (Borel measure), if $\int_{\mathcal{Y}} f \ d\mu_a = \int_{\mathcal{Y}} f \ d\mu_b$ for all $f \in C_b(\mathcal{Y})$; then $\mu_a = \mu_b$.

Combine (24) and (25), one can obtain $\mathcal{L}(\hat{T}, \hat{f}) - \mathcal{L}(\hat{T}, \nu\varphi) \le C(\rho_a, \rho_b) - \underline{c} - \nu\delta$. We can increase $\nu$ large enough so that $\mathcal{L}(\hat{T}, \hat{f}) < \mathcal{L}(\hat{T}, \nu\varphi)$. This contradiction to the fact that $\hat{f} \in \operatorname*{argmax}_{f \in C_b(\mathcal{Y})}\{\mathcal{L}(\hat{T}, f)\}$. Thus we have

proved that $\hat{T}_\sharp \rho_a = \rho_b$. As a result ,

$$\int_{\mathcal{X}} c(x, \hat{T}(x))\rho_a(x)dx = \mathcal{L}(\hat{T}, \hat{f}) = \inf_T\{\mathcal{L}(T, \hat{f})\} \le \inf_{T_\sharp \rho_a = \rho_b}\{\mathcal{L}(T, \hat{f})\} = \inf_{T_\sharp \rho_a = \rho_b}\int_{\mathcal{X}} c(x, T(x))\rho_a(x)dx.$$

This indicates that $\hat{T}$ solves the Monge problem (1). The above calculation also leads to $\mathcal{L}(\hat{T}, \hat{f}) = C_{\text{Monge}}(\rho_a, \rho_b)$.

At last, we prove that $\hat{f}$ is the optimal solution to Kantorovich dual problem (20). By applying Lemma 1, just notice that for any $f \in C_b(\mathcal{Y})$, one can verify

$$\mathcal{L}(\hat{T}, \hat{f}) \ge \mathcal{L}(\hat{T}, f) \ge \inf_T\{\mathcal{L}(T, f)\} = \int_{\mathcal{Y}} f(y)\rho_b(y)dy - \int_{\mathcal{X}} f^{c,-}(x)\rho_b(y)dy.$$

On the other hand, we have

$$\mathcal{L}(\hat{T}, \hat{f}) = \inf_T\{\mathcal{L}(T, \hat{f})\} = \int_{\mathcal{Y}} \hat{f}(y)\rho_b(y)dy - \int_{\mathcal{X}} \hat{f}^{c,-}(x)\rho_b(y)dy.$$

This proves that $\hat{f}$ is the optimal solution to (20). $\qquad\square$

A simialr proof of $T_* \in \operatorname*{argmin}_{T \in \mathcal{M}(\mathcal{X}, \mathcal{Y})}\{\mathcal{L}(T, \phi_*)\}$ as part of our proof to Theorem 2 is also established in section 6.1 of Gazdieva et al. (2022).

To prove the Corollary 1, it suffices to check the conditions in this corollary imply the conditions in Theorem 1, 2, 6.

## B.2 Proof of Theorem 3

*Proof .* As mentioned before in the proof of Theorem 2, by Lemma 1, one can verify

$$\inf_T \mathcal{L}(T, f) = \int_{\mathcal{Y}} f(y)\rho_b(y)dy - \int_{\mathcal{X}} f^{c,-}(x)\rho_a(x)dx, \tag{26}$$

Then the sup-inf problem $\sup_f \inf_T \mathcal{L}(T, f)$ can be formulated as

$$\sup_f \left\{\int_{\mathcal{Y}} f(y)\rho_b(y)dy - \int_{\mathcal{X}} f^{c,-}(x)\rho_a(x)dx\right\}.$$

This is exactly the Kantorovich dual problem (20). Since $(\bar{T}, \bar{f})$ is the solution to sup-inf problem, $\bar{f}$ is an optimal solution to (20). This verifies the first assertion of the theorem.

On the other hand, at $(\bar{T}, \bar{f})$, we have

$$\bar{T}(x) \in \operatorname{argmax}_{\xi \in \mathcal{Y}}\{\bar{f}(\xi) - c(x, \xi)\}, \ \rho_a \text{ almost surely.}$$

This leads to

$$\bar{f}^{c,-}(x) = \bar{f}(\bar{T}(x)) - c(x, \bar{T}(x)), \ \rho_a \text{ almost surely.}$$

Then we have

$$\begin{aligned}
\int_{\mathcal{X}} c(x, \bar{T}(x))\rho_a(x) \ dx &= \int_{\mathcal{X}} \bar{f}(\bar{T}(x))\rho_a(x) \ dx - \int_{\mathcal{X}} \bar{f}^{c,-}(x)\rho_a(x) \ dx \\
&= \int_{\mathcal{Y}} \bar{f}(y)\rho_b(y) \ dy - \int_{\mathcal{X}} \bar{f}^{c,-}(x)\rho_a(x) \ dx \\
&= \iint_{\mathcal{X} \times \mathcal{Y}} [\bar{f}(y) - \bar{f}^{c,-}(x)]d\pi(x, y) \le \iint_{\mathcal{X} \times \mathcal{Y}} c(x, y)d\pi(x, y)
\end{aligned}$$

for any $\pi \in \Pi(\rho_a, \rho_b)$. Here the second equality is due to the assumption $T_{*\sharp}\rho_a = \rho_b$. The last inequality is due to the definition of $\bar{f}^{c,-}(x) = \sup_y \{\bar{f}(y) - c(x,y)\}$.

We now take the infimum value of $\iint_{\mathcal{X} \times \mathcal{Y}} c d\pi$ and we obtain

$$\int_{\mathcal{X}} c(x, \bar{T}(x))\rho_a(x) \; dx \leq C(\rho_a, \rho_b), \tag{27}$$

Now, for any transport map $T$ satisfying $T_\sharp \rho_a = \rho_b$, by denoting $\pi = (\text{Id}, T)_\sharp \rho_a$, we have

$$\int_{\mathcal{X}} c(x, T(x))\rho_a(x) \; dx = \iint_{\mathcal{X} \times \mathcal{Y}} c(x,y) d\pi(x,y) \geq C(\rho_a, \rho_b). \tag{28}$$

Combining (27) and (28), we obtain

$$\int_{\mathcal{X}} c(x, \bar{T}(x))\rho_a(x) dx \leq \int_{\mathcal{X}} c(x, T(x))\rho_a(x) dx, \quad \text{for any } T, \; T_\sharp \rho_a = \rho_b.$$

This indicates the existence of the Monge map, and $\bar{T}$ is the Monge map.

At last, we have

$$\mathcal{L}(\bar{T}, \bar{f}) = \int_{\mathcal{X}} c(x, \bar{T}(x))\rho_a(x) dx + \int_{\mathcal{Y}} \bar{f}(y)(T_{*\sharp}\rho_a - \rho_b) dy = \int_{\mathcal{X}} c(x, \bar{T}(x))\rho_a(x) \; dx = C_{\text{Monge}}(\rho_a, \rho_b).$$

$\square$

### B.3 Consistency in optimal values

As discussed in Theorem 3, we require the condition $\bar{T}_\sharp \rho_a = \rho_b$ to establish the consistency between the sup-inf solution and the optimal transport solution. However, even without the assumption $\bar{T}_\sharp \rho_a = \rho_b$, one can directly prove the consistency in optimal value, i.e., $\sup_f \inf_T \mathcal{L}(T, f) = C(\rho_a, \rho_b)$. This is demonstrated in the following theorem.

**Theorem 7** (Consistency in optimal value). *Assume the cost function is continuous and satisfies the condition* (15). *Suppose the sup-inf problem* (4) *admits at least one solution* $(\bar{T}, \bar{f})$, *then* $\mathcal{L}(\bar{T}, \bar{f}) = C(\rho_a, \rho_b)$.

*Proof.* The proof of the first part of this theorem is the same as Theorem 3. Thus it is not hard to verify

$$\sup_f \inf_T \mathcal{L}(T, f) = \sup_f \left\{ \int_{\mathbb{R}^m} f(y)\rho_b(y) dy - \int_{\mathbb{R}^n} f^{c,-}(x)\rho_a(x) dx \right\} = K(\rho_a, \rho_b).$$

Since $c(x, y)$ satisfies (15), recall Theorem 6, the optimal value of Kantorovich dual problem equals general OT distance $C(\rho_a, \rho_b)$. This proves our assertion. $\square$

Theorem 7 indicates that although for some general cases in which our proposed method (4) fails to compute for a valid transport map $\bar{T}$, (4) is still able to recover the exact optimal transport distance $C(\rho_a, \rho_b)$. In the following example, although the Monge map does not exist, the proposed method can still capture the exact OT distance.

**Example 1.** *Consider the OT problem on $\mathbb{R}$ with $c(x, y) = |x - y|^2$ and $\rho_a = \delta_0$ (point distribution at 0), $\rho_b = \mathcal{N}(0, 1)$ (normal distribution). Our method yields*

$$\sup_f \inf_T \left\{ |0 - T(0)|^2 + \int_{\mathbb{R}} f(y)\rho_b(y) dy - f(T(0)) \right\}.$$

*By setting $\Psi_0(y) = f(y) - |y|^2$, our sup-inf problem becomes*

$$\sup_f \inf_T \left\{ \int_{\mathbb{R}} [\underbrace{(f(y) - |y|^2)}_{\Psi_0(y)} + |y|^2] \rho_b(y) dy - \underbrace{(f(T(0)) - |0 - T(0)|^2)}_{\Psi_0(T(0))} \right\}$$

$$= \sup_{\Psi_0} \inf_T \{ \int_{\mathbb{R}} (\Psi_0(y) + |y|^2) \rho_b(y) dy - \Psi_0(T(0)) \} = \underbrace{\int_{\mathbb{R}} |y|^2 \rho_b(y) dy}_{=1} + \sup_{\Psi_0} \underbrace{\left\{ \int_{\mathbb{R}} [\Psi_0(y) - \sup \Psi_0] \rho_b(y) dy \right\}}_{\leq 0} = 1$$

*The supreme over $\Psi_0$ is obtained when $\Psi_0 = Const$, i.e., when $f(y) = |y|^2 + Const$. Thus we have $\sup_f \inf_T \mathcal{L}(T, f) = 1 = C(\rho_a, \rho_b)$.*

### B.4 Proof of Theorem 4

We can firstly verify the following lemma.

**Lemma 2** (Existence and uniqueness of Monge map under Assumption 1, 2). *Suppose Assumption 1 and 2 hold, then the Monge map $T_*$ exists and is unique in law.*

Lemma 2 is a corollary of Theorem 1. Once assumption 1 and 2 hold, the conditions on $c(\cdot, \cdot)$ and $\rho_a, \rho_b$ mentioned in Theorem 1 are all satisfied. To prove Theorem 4, we further need the following two lemmas.

**Lemma 3.** *Suppose $n \times n$ matrix $A$ is invertible with minimum singular value $\sigma_{\min}(A) > 0$. Also assume $n \times n$ matrix $H$ is self-adjoint and satisfies $\lambda I_n \succeq H \succ O_n$[9]. Then $A^\top H^{-1} A \succeq \frac{\sigma_{\min}(A)^2}{\lambda} I_n$.*

*Proof of Lemma 3 .* One can first verify that $H^{-1} \succeq \frac{1}{\lambda} I_n$ by digonalizing $H^{-1}$. To prove this lemma, we only need to verify that for arbitrary $v \in \mathbb{R}^n$,

$$v^\top A^\top H^{-1} A v = (Av)^\top H^{-1} A v \geq \frac{|Av|^2}{\lambda} \geq \frac{\sigma_{\min}(A)^2}{\lambda} |v|^2$$

Thus $A^\top H^{-1} A - \frac{\sigma_{\min}(A)^2}{\lambda} I_n$ is positive-semidefinite. $\qquad\square$

The following lemma is crucial since it analyzes the concavity of the target function $f(\cdot) - c(\cdot, y)$ when $f$ is $c$-concave.

**Lemma 4** (Concavity of $f(\cdot) - c(x, \cdot)$ if $f$ $c$-concave). *Suppose the cost function $c(x, y)$ and $f$ satisfy the conditions mentioned in Theorem 4. Denote the function $\Psi_x(y) = f(y) - c(x, y)$, then we have*

$$\nabla^2 \Psi_x(y) \preceq -\frac{\sigma(x, y)^2}{\lambda(y)} I_n.$$

*Proof of Lemma 4.* First, we notice that $f$ is $c$-convex, thus, there exists $\varphi$ such that $f(y) = \inf_x \{\varphi(x) + c(x, y)\}$. Let us also denote $\Phi(x, y) = \varphi(x) + c(x, y)$.

Now for a fixed $y \in \mathbb{R}^n$, We pick one

$$x_y \in \operatorname{argmin}_x \{\varphi(x) + c(x, y)\}$$

Since we assumed that $\varphi \in C^2(\mathbb{R}^n)$ and $c \in C^2(\mathbb{R}^n \times \mathbb{R}^n)$, we have

$$\nabla_x \Phi(x_y, y) = \nabla \varphi(x_y) + \nabla_x c(x_y, y) = 0 \tag{29}$$

Now recall Assumption 3, $.\nabla^2_{xx} \Phi(x_y, y)$ is positive definite, thus is also invertible. We can now apply the implicit function theorem to show that the equation $\nabla_x \Phi(x, y) = 0$ determines an implicit function $x(\cdot)$,

---

[9] Here matrix $M_1 \succ M_2$ iff $M_2 - M_1$ is a positive-definite matrix, and $M_1 \succeq M_2$ iff $M_1 - M_2$ is a positive-semidefinite matrix.

which satisfies $x(y) = x_y$ in a small neighborhood $U \subset \mathbb{R}^n$ containing $y$. Furthermore, one can show that $x(\cdot)$ is continuously differentiable at $y$. We will denote $x_y$ as $x(y)$ in our following discussion.

Now differentiating (29) with respect to $y$ yields

$$\nabla^2_{xx}\Phi(x(y),y)\nabla x(y) + \nabla^2_{xy}c(x(y),y) = 0. \tag{30}$$

On one hand, (30) tells us

$$\nabla x(y) = -\nabla^2_{xx}\Phi(x(y),y)^{-1}\nabla^2_{xy}c(x(y),y). \tag{31}$$

Now we directly compute

$$\nabla^2\Psi_x(y) = \nabla^2 f(y) - \nabla^2_{yy}c(x,y). \tag{32}$$

In order to compute $\nabla^2 f(y)$, we first compute $\nabla f(y)$

$$\nabla f(y) = \nabla(\varphi(x(y)) + c(x(y),y)) = \nabla_y c(x(y),y). \tag{33}$$

the second equality is due to the envelope theorem Afriat (1971). Then $\nabla^2 f(y)$ can be computed as

$$\nabla^2 f(y) = \nabla^2_{yx}c(x(y),y)\nabla x(y) + \nabla^2_{yy}c(x(y),y). \tag{34}$$

Plugging (31) into (34), recall (32), this yields

$$\nabla^2\Psi_x(y) = -\nabla^2_{yx}c(x(y),y)\nabla^2_{xx}\Phi(x(y),y)^{-1}\nabla^2_{xy}c(x(y),y) + \nabla^2_{yy}c(x(y),y) - \nabla^2_{yy}c(x,y).$$

Recall the Assumption 1, notice that $c \in C^2(\mathbb{R}^n \times \mathbb{R}^n)$, by symmetry of second derivatives, one can verify $\nabla^2_{xy}c^\top = \nabla^2_{yx}c$; Since $\nabla^2_{yy}c(x,y)$ is independent of $x$, one has $\nabla^2_{yy}c(x(y),y) - \nabla^2_{yy}c(x,y) = 0$. Thus we obtain

$$\nabla^2\Psi_x(y) = -\nabla^2_{xy}c(x(y),y)^\top\nabla^2_{xx}\Phi(x(y),y)^{-1}\nabla^2_{xy}c(x(y),y). \tag{35}$$

By the positive definite assumption made in Assumption 3 as well as (7), we have $\lambda(y)I_n \succeq \nabla^2_{xx}\Phi(x(y),y) \succ O_n$. Recall that $\sigma_{\min}(\nabla^2_{xy}c(x,y)) = \sigma(x,y)$. Now we apply lemma 3 to (35), this yields

$$\nabla^2\Psi_x(y) \preceq -\frac{\sigma(x,y)^2}{\lambda(y)}I_n.$$

$\square$

Now we prove the main result of Theorem 4.

*Proof of Theorem 4.* In this proof, we denote $\int$ as $\int_{\mathbb{R}^d}$ for simplicity.

We first recall

$$\mathcal{L}(T,f) = \int c(x,T(x))\rho_a(x)\,dx + \int f(y)\rho_b(y)\,dy - \int f(T(x))\rho_a(x)\,dx$$
$$= \int f(y)\rho_b(y)\,dy - \int (f(T(x)) - c(x,T(x)))\rho_a(x)dx,$$

then we write

$$\mathcal{E}_1(T,f) = \mathcal{L}(T,f) - \inf_{\widetilde{T}}\mathcal{L}(\widetilde{T},f) = -\int[f(T(x)) - c(x,T(x))]\rho_a\,dx + \sup_{\widetilde{T}}\left\{\int[f(\widetilde{T}(x)) - c(x,\widetilde{T}(x))]\rho_a\,dx\right\}$$

Recall $T_f$ defined in Assumption 4, we have, for almost every $x \in \mathbb{R}^d$,

$$T_f(x) = \text{argmax}_y\{f(y) - c(x,y)\} = \text{argmax}_y\{\Psi_x(y)\}, \tag{36}$$

recall that we denote $\Psi_x(y) = f(y) - c(x, y)$, then, for almost every $x \in \mathbb{R}^d$, we have

$$\nabla \Psi_x(T_f(x)) = 0. \tag{37}$$

One can also write:

$$\mathcal{E}_1(T, f) = \int [(f(T_f(x)) - c(x, T_f(x))) - (f(T(x)) - c(x, T(x)))]$$
$$= \int [\Psi_x(T_f(x)) - \Psi_x(T(x))]\rho_a(x) \ dx$$

For any $x \in \mathbb{R}^d$, since $\Psi_x(\cdot) \in C^2(\mathbb{R}^n)$, and according to the previous Lemma 4, we have

$$\Psi_x(T(x)) - \Psi_x(T_f(x)) = \nabla \Psi_x(T_f(x))(T(x) - T_f(x)) + \frac{1}{2}(T(x) - T_f(x))^\top \nabla^2 \Psi_x(\omega(x))(T(x) - T_f(x))$$

with

$$\omega(x) = (1 - \theta_x)T(x) + \theta_x T_f(x) \tag{38}$$

for certain $\theta_x \in [0, 1]$. By (37) and Lemma 4, we have

$$\Psi_x(T(x)) - \Psi_x(T_f(x)) \leq -\frac{\sigma(x, \omega(x))^2}{2\lambda(\omega(x))}|T(x) - T_f(x)|^2.$$

Thus we have:

$$\mathcal{E}_1(T, f) = \int [\Psi_x(T_f(x)) - \Psi_x(T(x))]\rho_a(x) \ dx \geq \int \frac{\sigma(x, \omega(x))^2}{2\lambda(\omega(x))}|T(x) - T_f(x)|^2\rho_a(x) \ dx \tag{39}$$

On the other hand, recall a solution of our sup-inf problem (4) is denoted as $(\bar{T}, \bar{f})$. Then we have

$$\sup_f \inf_T \mathcal{L}(T, f) = \int c(x, \bar{T}(x))\rho_a \ dx - \int f(y)(\bar{T}_\sharp \rho_a - \rho_b)dy = \int c(x, \bar{T}(x))\rho_a \ dx,$$

the second equality is due to the assumption on the consistency between $\bar{T}$ and Monge map $T_*$, thus $\bar{T}_\sharp \rho_a = \rho_b$. Thus we have

$$\mathcal{E}_2(f) = \int c(x, \bar{T}(x))\rho_a \ dx - \inf_{\widetilde{T}} \left( \int c(x, \widetilde{T}(x))\rho_a(x)dx + \int f(y)\rho_b(y) \ dy - \int f(\widetilde{T}(x))\rho_a(x)dx \right)$$
$$= -\int f(\bar{T}(x)) - c(x, \bar{T}(x))\rho_a \ dx + \sup_{\widetilde{T}} \int (f(\widetilde{T}(x)) - c(x, \widetilde{T}(x)))\rho_a(x)dx.$$

The second equality is due to $\bar{T}_\sharp \rho_a = \rho_b$. Similar to the previous treatment, we have

$$\mathcal{E}_2(f) = \int [\Psi_x(T_f(x)) - \Psi_x(\bar{T}(x))]\rho_a(x) \ dx$$

Apply similar analysis as before, we obtain

$$\mathcal{E}_2(f) \geq \int \frac{\sigma(x, \xi(x))^2}{2\lambda(\xi(x))}|\bar{T}(x) - T_f(x)|^2\rho_a(x) \ dx \tag{40}$$

with

$$\xi(x) = (1 - \tau_x)\bar{T}(x) + \tau_x T_f(x) \tag{41}$$

for certain $\tau_x \in [0, 1]$. Since $\bar{T} = T_*$, $\rho_a$ almost surely, (40) leads to

$$\mathcal{E}_2(f) \geq \int \frac{\sigma(x, \xi(x))^2}{2\lambda(\xi(x))} |T_*(x) - T_f(x)|^2 \rho_a(x) \ dx \tag{42}$$

Now we set

$$\beta(x) = \min \left\{ \frac{\sigma(x, \omega(x))^2}{2\lambda(\omega(x))}, \frac{\sigma(x, \xi(x))^2}{2\lambda(\xi(x))} \right\}, \tag{43}$$

where $\omega$ and $\xi$ are defined in (38) and (41). Combining (39) and (42), we obtain

$$\mathcal{E}_1(T, f) + \mathcal{E}_2(f) \geq \int \beta(x)(|T(x) - T_f(x)|^2 + |T_*(x) - T_f(x)|^2)\rho_a \ dx$$

$$\geq \int \frac{\beta(x)}{2} |T(x) - T_*(x)|^2 \rho_a \ dx$$

This leads to $\|T - T_*\|_{L^2(\beta\rho_a)} \leq \sqrt{2(\mathcal{E}_1(T, f) + \mathcal{E}_2(f))}$.

$\square$

**Discussion on enforcing the c-concave property of the dual variable**  Theorem 4 is established on the assumption of $c$-concavity of the dual variable $f$. In practice, the $c$-concave property can be approximated via soft-min convolution, i.e., one can set $f_\theta(y) = -\frac{1}{K} \log(\mathbb{E}_{\boldsymbol{X} \sim U} \exp(-K(\varphi_\theta(\boldsymbol{X}) + c(\boldsymbol{X}, y))))$ with $K > 0$ large enough, $\varphi_\theta$ as an arbitrary neural network and $U$ as a uniform distribution over a sufficiently large ball. However, according to the existing work (Fan et al., 2022b; Korotin et al., 2022a) on applying ICNN to solve OT, restricting $f$ to the $c$-concave functions may undermine the performance of our computation. The related implementation and comparison may serve as good directions for future research.

### B.5   Further analysis on $\lambda(\cdot)$ and $\beta(\cdot)$

We can carry out some further calculations so that we may gain a better understanding of the value $\lambda(y)$ evaluated at $y = T_f(x)$. We can then tell that the quantity $\beta$ defined in (43) will be impossible to be bounded from below as the maps $T, T_f$ are very close to the sup-inf solution $\bar{T}$. Before our calculation, we need the following lemma.

**Lemma 5.** *We assume the assumptions mentioned in Theorem 4 all hold. We also assume that for the cost function $c$, the map $\nabla_y c(\cdot, y) : x \mapsto \nabla_y c(x, y)$ is injective. We further strengthen Assumption 4 such that $T_f(x)$ minimizes $f(y) - c(x, y)$ for all $x \in \mathbb{R}^d$. Similarly, we denote map $T_\varphi$ as $T_\varphi(y) = x(y) \in \arg\min_x \{\varphi(x) + c(x, y)\}$ for all $y \in \mathbb{R}^d$. Then we have $T_\varphi(T_f(y)) = y$ for all $y$.*

*Proof.* To prove this, we notice that $\nabla_y(f(y) - c(x, y))|_{y=T_f(x)} = 0$. This yields

$$\nabla f(T_f(x)) - \nabla_y c(x, T_f(x)) = 0, \quad \text{for arbitrary } x \in \mathbb{R}^d. \tag{44}$$

By a similar argument carried out in the proof of Lemma 4, we are able to show that

$$\nabla f(y) = \nabla_y c(T_\varphi(y), y), \tag{45}$$

for all $y \in \mathbb{R}^d$. Then, for any $x$, we can set $y = T_f(x)$ in (45) and plug this into (44) to obtain

$$\nabla_y c(T_\varphi(T_f(x)), T_f(x)) - \nabla_y c(x, T_f(x)) = 0 \tag{46}$$

Now due to the fact that $\nabla_{xy} c(x, y)$ is invertible in Assumption 1, one can tell that the map $\nabla_y c(\cdot, y) : x \mapsto \nabla_x c(x, y)$ is injective. Thus (46) yields $T_\varphi(T_f(x)) = x$ for all $x \in \mathbb{R}^d$. $\square$

**Remark 6** ($T_f, T_\varphi$ as inverse map to each other). *If we further assume that $\varphi$ is c-convex, i.e., $\varphi(x) = \sup_y \{f(y) - c(x, y)\}$. Then by similar argument, we can also prove that $T_f(T_\varphi(y)) = y$ for all $y \in \mathbb{R}^d$. Thus, $T_f$ and $T_\varphi$ are inverse map to each other.*

**Theorem 8** (Estimation on $\lambda(\cdot)$ at $T_f(x)$). *Suppose we assume all the assumptions as in Lemma 5. Let $T_f$ as defined in Assumption 4, and $\lambda(\cdot)$ defined in (7). Denote $\Gamma = \nabla^2_{xy}c(x, T_f(x))$. Then we have $\lambda(T_f(x)) = \|\Gamma DT_f(x)\Gamma^{-1}\Gamma^\top\|_2$. Furthermore, $\lambda(T_f(x)) \geq \sigma(\Gamma)\|DT_f(x)\|_2$.*

*Proof.* Let us recall that we denote $T_f(x) = \arg\max_y\{f(y) - c(x,y)\} = \arg\max_y\{\Psi_x(y)\}$. Using the optimality condition, we obtain $\nabla\Psi_x(T_f(x)) = 0$.

$$\nabla_y c(x, T_f(x)) - \nabla f(T_f(x)) = 0. \tag{47}$$

Let us now differentiate (47) w.r.t. $x$.

$$\nabla^2_{xy}c(x, T_f(x)) + \nabla^2_{yy}c(x, T_f(x))DT_f(x) - \nabla^2 f(T_f(x))DT_f(x) = 0,$$

which can also be formulated as

$$\nabla^2_{xy}c(x, T_f(x)) + \nabla^2_{yy}\Psi_x(T_f(x))DT_f(x) = 0. \tag{48}$$

Let us now recall equation (35) and the definition of $T_\varphi$, we substitute (35) into (48) to obtain

$$\nabla^2_{xy}c(x, T_f(x)) - \nabla^2_{xy}c(T_\varphi(T_f(x)), T_f(x))^\top \nabla^2_{xx}\Phi(T_\varphi(T_f(x)), T_f(x))^{-1}\nabla^2_{xy}c(T_\varphi(T_f(x)), T_f(x))DT_f(x) = 0. \tag{49}$$

Recall that the previous Lemma 5 yields $T_\varphi(T_f(x)) = x$ for all $x \in \mathbb{R}^d$. Let us denote $\Gamma = \nabla^2_{xy}c(x, T_f(x))$ for simplicity. Then equation (49) then leads to

$$\nabla^2_{xx}\Phi(x(T_f(x)), T_f(x)) = \Gamma DT_f(x)\Gamma^{-1}\Gamma^\top \tag{50}$$

Since $\nabla^2_{xx}\Phi$ is self-adjoint, then $\lambda_{\max}(\nabla^2_{xx}\Phi(x(T_f(x)), T_f(x))) = \|\nabla^2_{xx}\Phi(x(T_f(x)), T_f(x))\|_2 = \|\Gamma DT_f(x)\Gamma^{-1}\Gamma^\top\|_2$. We can bound this matrix 2-norm as

$$\|\Gamma DT_f(x)\Gamma^{-1}\Gamma^\top\|_2 \geq \frac{\|\Gamma DT_f(x)\Gamma^{-1}\|_2}{\|\Gamma^{-\top}\|_2} = \sigma_{\min}(\Gamma)\|DT_f(x)\|_2.$$

Thus, when $y$ equals to the minimizer $T_f(x) \in \arg\min_y\{f(y) - c(x,y)\}$, the quantity $\lambda(y) = \lambda(T_f(x)) \geq \sigma_{\min}(\Gamma)\|DT_f(x)\|_2$ $\square$

Now suppose that method is about to converge to the sup-inf solution, i.e., the current minimizer $T_f$ of $\mathcal{L}(f, T)$ is very close to the sup-inf solution $\bar{T}$ and our computed $T$ is very close to $T_f$. Then the $\omega(x), \xi(x)$ mentioned in the proof of Theorem 4 will both be very close to $T_f(x)$, thus, $\beta(x)$ approximately equals to

$$\frac{\sigma(x, T_f(x))}{\lambda(T_f(x))}$$

Recall the inequality $\lambda(T_f(x)) \geq \sigma(\Gamma)\|DT_f(x)\|_2$ by Theorem 8 and $\sigma_{\min}(\Gamma) = \sigma_{\min}(\nabla^2_{xy}c(x, T_f(x))) = \sigma(x, T_f(x))$ by definition, we obtain that $\beta(x)$ can approximately bounded from above by $\frac{1}{\|DT_f(x)\|_2}$. Notice that we can easily construct $\rho_a$ with very small support and $\rho_b$ with large support so that the norm of the gradient of the approximated OT map $\|T_f(x)\|_2$ is as large as possible for all $x \in \text{supp}(\rho_a)$. Then $\beta(x)$ can be an arbitrarily small value.

**Example 2.** *Consider quadratic cost $c(x,y) = |x-y|^2$ and two uniform distributions on $\mathbb{R}$, $\rho_a = U([-\epsilon, \epsilon])$ and $\rho_b = U([-L, L])$ with $\epsilon, L > 0$. Then the optimal transport map $T_*(x) = \frac{L}{\epsilon}x$. Since we assume the consistency, $\bar{T}(x) = \frac{L}{\epsilon}x$, thus $\|D\bar{T}\| = \frac{L}{\epsilon}$.*

In Example 2, suppose $T, T_f$ are very close to $\bar{T}$, it is likely that $DT_f(x)$ also possesses a large norm for most of the $x \in [-\epsilon, \epsilon]$, which leads to a very small value of $\beta(x) \leq \frac{\epsilon}{L}$.

## C  Relation between our method and generative adversarial networks

It is worth pointing out that our scheme and Wasserstein Generative Adversarial Networks (WGAN) (Arjovsky et al., 2017) are similar in the sense that they are both doing minimization over the generator/map and maximization over the discriminator/dual potential. However, there are two main distinctions between them. Such differences are not reflected from the superficial aspects such as the choice of reference distributions $\rho_a$, but come from the fundamental logic hidden behind the algorithms.

- We want to first emphasize that the mechanisms of the two algorithms are different: Typical Wasserstein GANs (WGAN) are usually formulated as

$$\min_{G} \underbrace{\max_{\|D\|_{\mathrm{Lip} \leq 1}} \int D(y)\rho_b(y)dy - \int D(G(x))\rho_a(x)dx}_{1-\mathrm{Wasserstein\ distance}\ W_1(G_\sharp \rho_a, \rho_b)} \tag{51}$$

and ours reads

$$\max_{f} \min_{T} \underbrace{\int f(y)\rho_b(y)dy - \int f(T(x))\rho_a(x)dx + \int c(X, T(x))\rho_a(x)dx}_{\mathrm{general\ Wasserstein\ distance}\ C_{\mathrm{Monge}}(\rho_a, \rho_b)} \tag{52}$$

The inner maximization of (51) computes $W_1$ distance via Kantorovich duality and the outer loop minimize the $W_1$ gap between desired $\rho_b$ and $G_\sharp \rho_a$; However, the logic behind our scheme (52) is different: the inner optimization computes for the $c-$transform of $f$, i.e. $f^{c,-}(x) = \sup_{\xi}(f(\xi) - c(x, \xi))$; And the outer maximization computes for the Kantorovich dual problem $C(\rho_a, \rho_b) = \sup_f \left\{ \int f(y)\rho_b(y)dy - \int f^{c,-}(x)\rho_a(x)dx \right\}$.

Even under $W_1$ circumstance, one can verify the intrinsic difference between two proposed methods: when setting the cost $c(x, y) = \|x - y\|$, and $\rho_a = G_\sharp \rho_a$ in (52), the entire "sup-inf" optimization of (52) (underbraced part) is equivalent to the inner maximization problem of (51) (underbraced part), but not for the entire saddle point scheme.

It is also important to note that WGAN aims to minimize the distance between the generated distribution and the target distribution and the ideal value for (51) is 0. On the other hand, one of our goals is to estimate the optimal transport distance between the initial distribution $\rho_a$ and the target distribution $\rho_b$. Thus the ideal value for (52) should be $C(\rho_a, \rho_b)$, which is not 0 in most of the cases.

- We then argue about the optimality of the computed map $G$ and $T$: In (51), one is trying to obtain a map $G$ by minimizing $W_1(\rho_b, G_\sharp \rho_a)$ w.r.t. $G$, and hopefully, $G_\sharp \rho_a$ can approximate $\rho_b$ well. However, there isn't any restriction exerted on $G$, thus one can not expect the computed $G$ to be the optimal transport map between $\rho_a$ and $\rho_b$; On the other hand, in (52), we not only compute $T$ such that $T_\sharp \rho_a$ approximates $\rho_b$, but also compute for the optimal $T$ that minimizes the transport cost $\mathbb{E}_{\rho_a}[c(X, T(X))]$. In (52), the computation of $T$ is naturally incorporated in the sup-inf scheme, and there exists the theoretical result (recall Theorem 3 in the paper) that guarantees $T$ to be the optimal transport map.

In summary, even though the formulation of both algorithms are similar, the designing logic (minimizing distance vs computing distance itself) and the purposes (computing arbitrary pushforward map vs computing the optimal map) of the two methods are distinct. Thus the theoretical and empirical study of GANs cannot be trivially translated to the proposed method. In addition to the above discussions, we should also refer the readers to Gazdieva et al. (2022), in which a comparison between a similar saddle point method and the regularized GANs are made in section 6.2 and summarized in Table 1.

## D  Additional results

**OT between 1D manifolds**  We consider the source distribution to be an 1D uniform distribution, and the target as an incomplete ellipse. In this case, the source dimension $n = 1$ and target dimension $m = 2$.

As discussed in Section 3, we choose the embedding function to be padding-zero function $Q(x) = [x, 0]$, so the cost reads

$$c(x, y) = \bar{c}(Q(x), y) = \|[x, 0] - y\|_2^2.$$

As shown in Figure 8, our algorithm is able to learn a symmetric map from an 1D uniform distribution towards the incomplete ellipse. This example also shows that our method can deal with the case that both $\rho_a$ and $\rho_b$ are not absolute continuous w.r.t. Lebesgue measure.

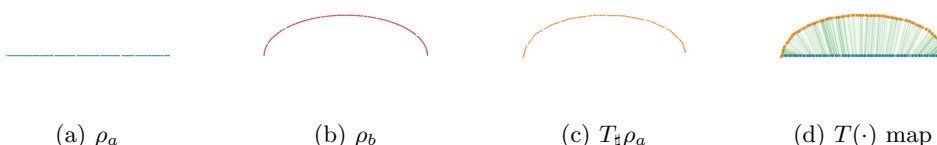

(a) $\rho_a$          (b) $\rho_b$          (c) $T_\sharp \rho_a$          (d) $T(\cdot)$ map

Figure 8: Qualitative results for learning unequal dimension maps.

**Decreasing function as the cost** We consider the cost function $c(x, y) = \phi(\|x - y\|)$ with $\phi$ as a monotonic decreasing function. We test our algorithm for a specific example $\phi(s) = \frac{1}{s^2}$. In this example, we compute for the optimal Monge map from $\rho_a$ to $\rho_b$ with $\rho_a = h_\sharp U([4, 6] \times [0, 2\pi))$ and $\rho_b = h_\sharp U([1, 2] \times [0, 2\pi))$. Here we define $h : \mathbb{R}_+ \times [0, 2\pi) \to \mathbb{R}^2$ as the transform from polar coordinates to Cartesian coordinates, i.e. $h(r, \theta) = (r \cos \theta, r \sin \theta)$. We denote $U(E)$ as the uniform distribution over the measurable set $E \subset \mathbb{R}^2$. Let us denote the outer annulus as $\Omega_a$, which is the image of $[4, 6] \times [0, 2\pi)$ under $h$; similarly, the inner annulus is denoted as $\Omega_b$. One can verify that the density function $\rho_a(x) = \frac{1}{4\pi\|x\|} \chi_{\Omega_a}$ and $\rho_b(x) = \frac{1}{2\pi\|x\|} \chi_{\Omega_b}$. Here $\chi_E$ is the indicator function of measurable set $E$, i.e. $\chi_E(x) = 1$ if $x \in E$ and $\chi_E(x) = 0$ if $x \notin E$.

Under polar coordinate, we define the map $\tau_*$ that maps each point $(r, \theta)$ on $[4, 6] \times [0, 2\pi)$ to $(4 - \frac{r}{2}, (\theta + \pi) \mod (2\pi)) \in [1, 2] \times [0, 2\pi)$. Then the ground truth Monge map is $T_* = h \circ \tau_* \circ h^{-1} : \Omega_a \to \Omega_b$. Now we denote OT distance using the above reciprocal cost as $W_{-2}$. Then it is not hard to compute $W_{-2}(\rho_a, \rho_b) = \iint_{\Omega_a} \frac{1}{\|T_*(x) - x\|^2} \rho_a(x) \, dx = \int_0^{2\pi} \int_4^6 \frac{1}{\left(4 + \frac{r}{2}\right)^2} \frac{1}{4\pi r} r \, dr \, d\theta = \frac{1}{2} \int_4^6 \frac{1}{\left(4 + \frac{r}{2}\right)^2} dr = \frac{1}{42}$.

We also compute the same problem for $L^2$ cost, i.e. $c(x, y) = \|x - y\|^2$. Under polar coordinate, the corresponding Monge map is $T_* = h \circ \tau_* \circ h^{-1}$ where $\tau_*$ of the problem should map each point $(r, \theta)$ on $[4, 6] \times [0, 2\pi)$ to $(\frac{r}{2} - 1, \theta)$. Similar to the above discussion, $W_2(\rho_a, \rho_b) = \left(\int_0^{2\pi} \int_4^6 \left(1 + \frac{r}{2}\right)^2 \frac{1}{4\pi r} r \, dr \, d\theta\right)^{\frac{1}{2}} = \sqrt{\frac{37}{3}}$

Figure 9 shows the transported samples as well as the differences between the two cost functions.

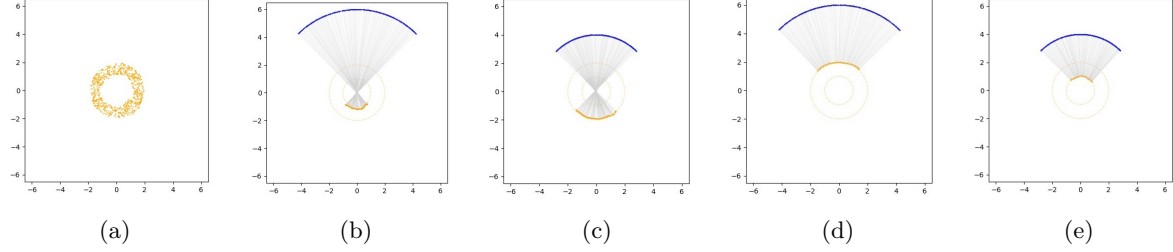

(a)        (b)        (c)        (d)        (e)

Figure 9: Panel (a) shows the samples from computed $T_\sharp \rho_a$. When $c(x, y) = 1/\|x - y\|^2$ (panels (b) (c)), we show $T(x)$ maps the circle with radius 6 to the circle with radius 1, and maps the circle with radius 4 to the circle with radius 2. If $c(x, y) = \|x - y\|^2$, which is the case of the panels (d) (e), it is exactly opposite.

**Uniform distribution on sphere** We use the geodesic distance on sphere (13) as the cost. We set $\rho_a = U([0, 2\pi] \times [0, \frac{\pi}{4}])$ and $\rho_b = U([0, 2\pi] \times [\frac{3\pi}{4}, \pi])$, and consider solving

$$\min_{T:T_\sharp\rho_a=\rho_b} \int c((\theta, \phi), T(\theta, \phi))\rho_a \; d\theta d\phi. \tag{53}$$

To mitigate the gradient blow-up of arccos near $\pm 1$, we slightly modify the original cost function as

$$c_\lambda((\theta_1, \phi_1), (\theta_2, \phi_2)) = \arccos(\lambda(\sin\phi_1 \sin\phi_2 \cos(\theta_1 - \theta_2) + \cos\phi_1 \cos\phi_2)) \tag{54}$$

with $\lambda = 0.999$. We apply our algorithm to solve (53) with cost $c_\lambda$. We also translate our computed map back to the 3D sphere $S$ via spherical-Cartesian coordinate transform $(\theta, \phi, r) \mapsto (r\sin\phi\cos\theta, r\sin\phi\sin\theta, r\cos\phi)$ where we fix the radius $r = 3$.

We demonstrate the numerical results in Figure 10. In this example, the ground truth Monge map $T_*$ of (53) maps each $(\theta, \phi)$ on $[0, 2\pi] \times [0, \frac{\pi}{4}]$ to $(\theta, \phi + \frac{3}{4}\pi)$. Furthermore, it is not hard to verify that the OT distance between $\rho_a, \rho_b$ is $W_{\text{geodesic}}(\rho_a, \rho_b) = \frac{9\pi}{4}$.

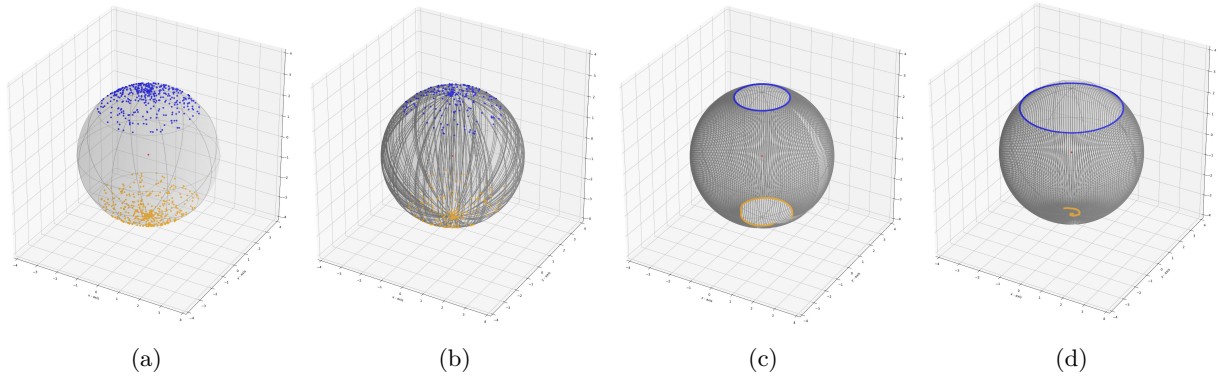

$$\text{(a)} \qquad\qquad \text{(b)} \qquad\qquad \text{(c)} \qquad\qquad \text{(d)}$$

Figure 10: Monge map from $\rho_a$ to $\rho_b$ on the sphere: (a) blue samples from $\rho_a$ and orange samples from $\rho_b$; (b) blue samples from $\rho_a$, orange samples are obtained from $T_\sharp\rho_b$, grey curves are geodesics connecting each transporting pairs; (c) our computed Monge map maps blue ring ($\phi = \frac{\pi}{8}$) to the orange curve (ground truth is $\phi = \frac{7}{8}\pi$); (d) our computed Monge map maps blue ring ($\phi = \frac{\pi}{4}$) to the orange curve (ground truth is the south pole)

Our computational method also yields to an estimation on the OT distance. To be more specific, for any computed $T_\theta$, we can use Monte Carlo method to estimate the OT distance as $\widehat{W}(\rho_a, \rho_b) = \frac{1}{N} \sum_{k=1}^{N} c(x_k, T_\theta(x_k))$, where the $N$ samples $\{x_k\}$ are i.i.d. samples drawn from $\rho_a$. In exact computation, we choose $N = 50000$. We summarize the comparison between our estimated OT distances and exact OT distances of previous examples in the following Table 3.

| | $W_{-2}$ (annulus example) Figure 9 (b,c) | $W_2$ (annulus example) Figure 9 (d,e) | $W_{\text{geodesic}}$ (sphere example) Figure 10 |
|---|---|---|---|
| real distance $W$ | $1/42 \approx 0.0238$ | $\sqrt{37/3} \approx 3.5119$ | $9\pi/4 \approx 7.0686$ |
| our estimation $\widehat{W}$ | 0.0241 | 3.4019 | 7.1041 |
| relative error $\frac{|W-\widehat{W}|}{|W|}$ | 1.3149% | 3.1325% | 0.5018% |

Table 3: Comparison between the estimated OT distances and the exact OT distances.

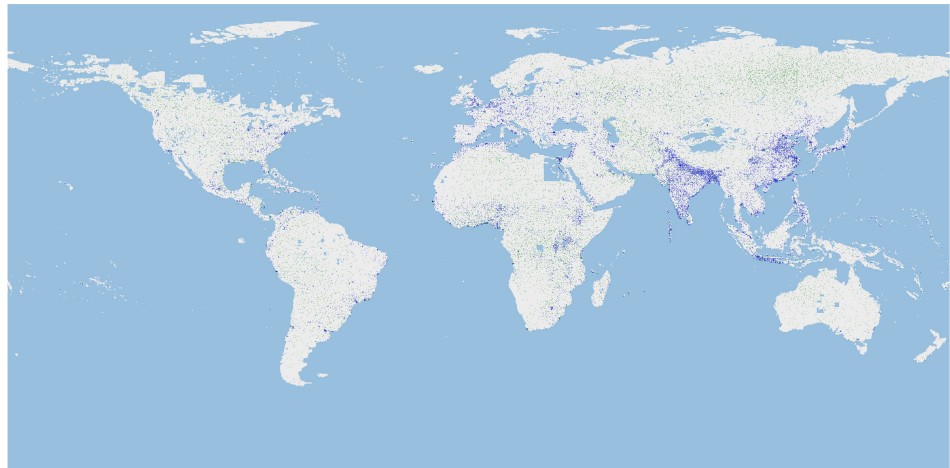

Figure 11: In this figure, we plot $N = 40000$ samples $\{\theta_k, \phi_k\}_{k=1}^N$ (blue) randomly drawn from $\rho_a^{\mathrm{Sph}}$, and their mapped samples $\{T_\theta(\theta_k, \phi_k)\}_{k=1}^N$ (green).

**Population transportation**   We have described this example in Section 6.4. Here we make further explanation on the map-to-land transform $\tau$, it is defined as follows.

$$\tau(\theta, \phi) = \begin{cases} (\theta, \phi), & \text{if } (\theta, \phi) \in \text{Land}; \\ \underset{(\tilde{\theta}, \tilde{\phi}) \in P}{\operatorname{argmin}} \left\{ \|(\tilde{\theta}, \tilde{\phi}) - (\theta, \phi)\|_2 \right\}, & \text{if } (\theta, \phi) \in \text{Sea}. \end{cases}$$

Here we choose $P$ as a finite set consists of 2000 samples randomly selected from $\rho_b^{\mathrm{Sph}}$.

We further plot in Figure 11 with sufficiently large amount of source samples and mapped samples. Among the 40000 mapped samples, 7718 are located on the sea, and we apply $\tau$ to map these samples back to a rather close location on land. It is worth mentioning that the map used as the background in our figures has several small regions removed from the actual land. This is due to the lack of data points in the dataset provided in Doxsey-Whitfield et al. (2015).

We also tested our algorithm on this population transport example with different costs. One of the costs we consider is the $c_\lambda$ cost defined before in (54), with $\lambda = 0.99$. We can treat $c_\lambda$ as a reasonable approximation of the exact geodesic distance on sphere. The other cost we consider is the $L^2$ cost $c((\theta_1, \phi_1), (\theta_2, \phi_2)) = (\theta_2 - \theta_1)^2 + (\phi_2 - \phi_1)^2$. We present the numerical results for both costs in Figure 12. One can tell that there are more transport trajectories passing through the Northern Arctic Circle with $c_\lambda$ cost than the trajectories with $L^2$ cost. This makes sense since the $c_\lambda$ distance of trajectory passing through the North Arctic Circle won't be necessarily large when comparing with $L^2$ distance. As a result, the transport plan with $c_\lambda$ cost is available to accommodate more transport plans going across the Arctic Circle.

**Text to image generation**   We show additional generation results in Figure 13. We also compared the cos-sim cost (9) and the ordinary inner product cost $c(x, y) = -\langle Rx, y \rangle$. The latter can converge to the same level as cos-sim cost, measured both qualitatively via generated images and quantitatively via cosine similarity w.r.t. real images (see Figure 4 (b).) The only difference is the method with inner product cost will converge slightly slower. After investigating the statistics of $\|Rx\|$, we find this result is reasonable because $\|Rx\|$ concentrates around 12 with a large probability. Meanwhile, we emphasize that this text-to-image generation task has not been considered in previous neural OT methods, which is itself a contribution, even though using cos-sim cost does not distinguish from inner product cost significantly.

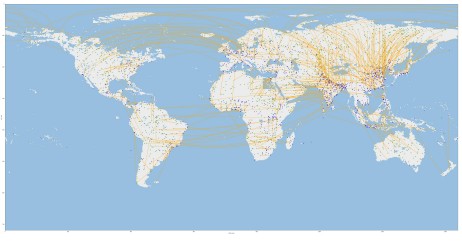 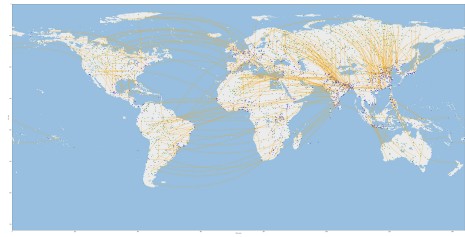

(a) Population transport with cost $c_\lambda$          (b) Population transport with $L^2$ cost

Figure 12: Comparison of transport plans using different cost functions.

## E   Implementation details and hyper-parameters

### E.1   Synthetic datasets

We follow Algorithm 1 and apply Adam method Kingma & Ba (2014) to train both $T_\theta$ and $f_\eta$ for all the following examples in this section. Recall the notations $B, K, K_1, K_2$ mentioned in Algorithm 1.

**OT between 1D manifolds**   The networks $T_\theta$ and $f_\eta$ each has 5 layers with 10 hidden neurons. The batch size $B = 100$. $K_1 = 6, K_2 = 1$. The learning rate is $10^{-3}$. The number of iterations $K = 12000$.

**Decreasing cost function**   In this example, we set $T_\theta(x) = x + F_\theta(x)$ and optimize over $\theta$. For either $\frac{1}{|x-y|^2}$ or $|x-y|^2$ case we set both $F_\theta$ and the Lagrange multiplier $f_\eta$ as six layers multilayer perceptron (MLP), with PReLU and Tanh activation functions respectively, each layer has 36 nodes. The training batch size $B = 2000$. We set $K = 8000, K_1 = 8, K_2 = 6$. The learning rate for both $\theta$ and $\eta$ equals $10^{-4}$.

**Uniform distribution on sphere**   In this example, we set $T_\theta(x) = x + F_\theta(x)$ and optimize over $\theta$. We set both $F_\theta$ and $f_\eta$ as six layers MLP, with PReLU activation functions, each layer has 8 nodes The training batch size is $B = 4000$. We set $K = 10000, K_1 = 8, K_2 = 4$. The learning rate for both $\theta$ and $\eta$ equals $0.9 \times 10^{-4}$.

**Population transportation**   In this example, we set $T_\theta(x) = x + F_\theta(x)$ and optimize over $\theta$, we choose both $F_\theta, f_\eta$ as ResNets with depth equals 4 and hidden dimensions equals 32; we choose the activation function as PReLU. We apply dropout technique Srivastava et al. (2014) with $p = 0.24$ to each layer of our networks. The learning rate for both $\theta$ and $\eta$ was set as $5 \times 10^{-5}$. $T_\theta$ is computed after $K = 200000$ steps of optimization with $K_1 = 8, K_2 = 1$ and training batchsize $B = 400$.

For the comparison experiment, we tested our example by slightly modifying the codes presented in OT mapping estimation for domain adaptation of POT library Flamary et al. (2021). We train the linear transformation on 2000 samples from both $\rho_a^{\text{Sph}}$ and $\rho_b^{\text{Sph}}$, we choose the hyper parameter $\mu = 1, \epsilon = 10^{-4}$. We plot the second Figure 7 by applying the map-to-land transform $\tau$ with 2000 newly selected samples. We also tried the same example with Gaussian kernel provided in Flamary et al. (2021), but the results are always unstable for various hyper parameters. Generally speaking, it is very hard to obtain valid results with Gaussian kernel.

### E.2   Text to image generation

**Dataset details**   Laion aesthetic dataset is filtered from a Laion 5B dataset to have the high aesthetic level. Laion art is the subset of Laion aesthetic and contains 8M most aethetic samples. We download the metadata of Laion art according to the instructions of Laion. We then filter only English prompts, and download the images with img2dataset. To speed up the training, we run the CLIP retrieval to convert

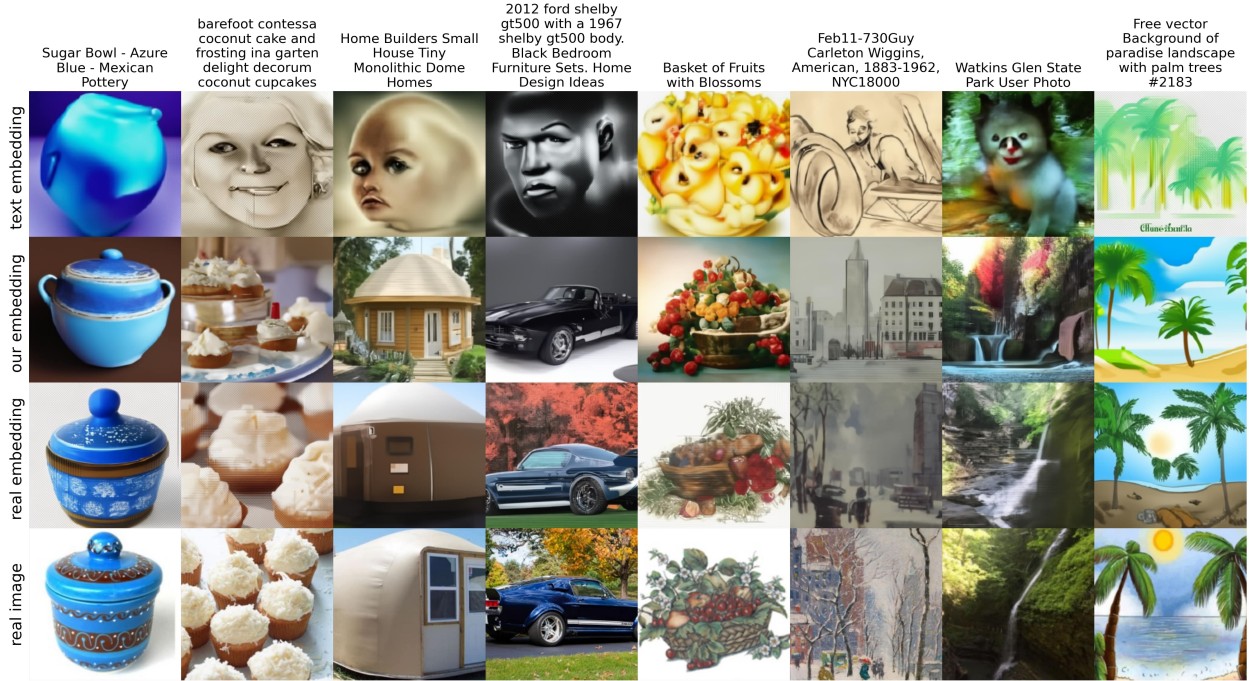

(a) Random image samples on Laion art prompts

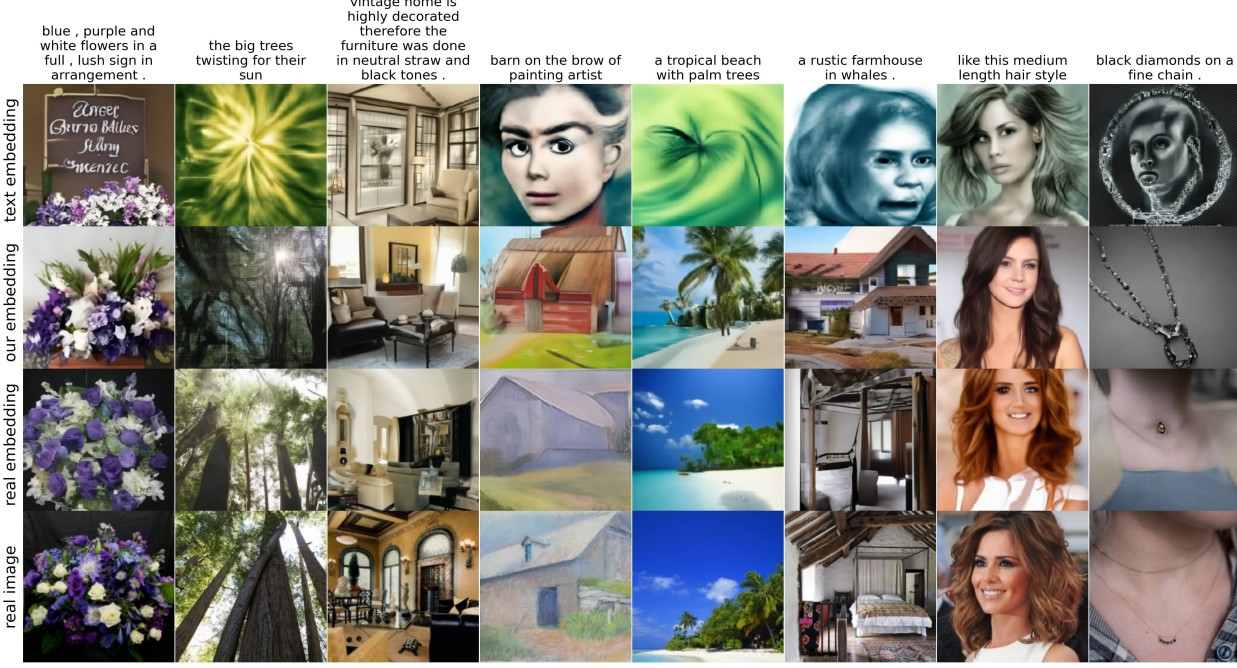

(b) Random image samples on Conceptual Captions 3M prompts

Figure 13: Additional text to image generation results

images to embeddings and use embedding-dataset-reordering to reorder the embeddings into the expected format. By doing these two steps, we save the time of calculating embeddings on the fly. After filtering English prompts, we get the Laion art dataset with 2.2M data.

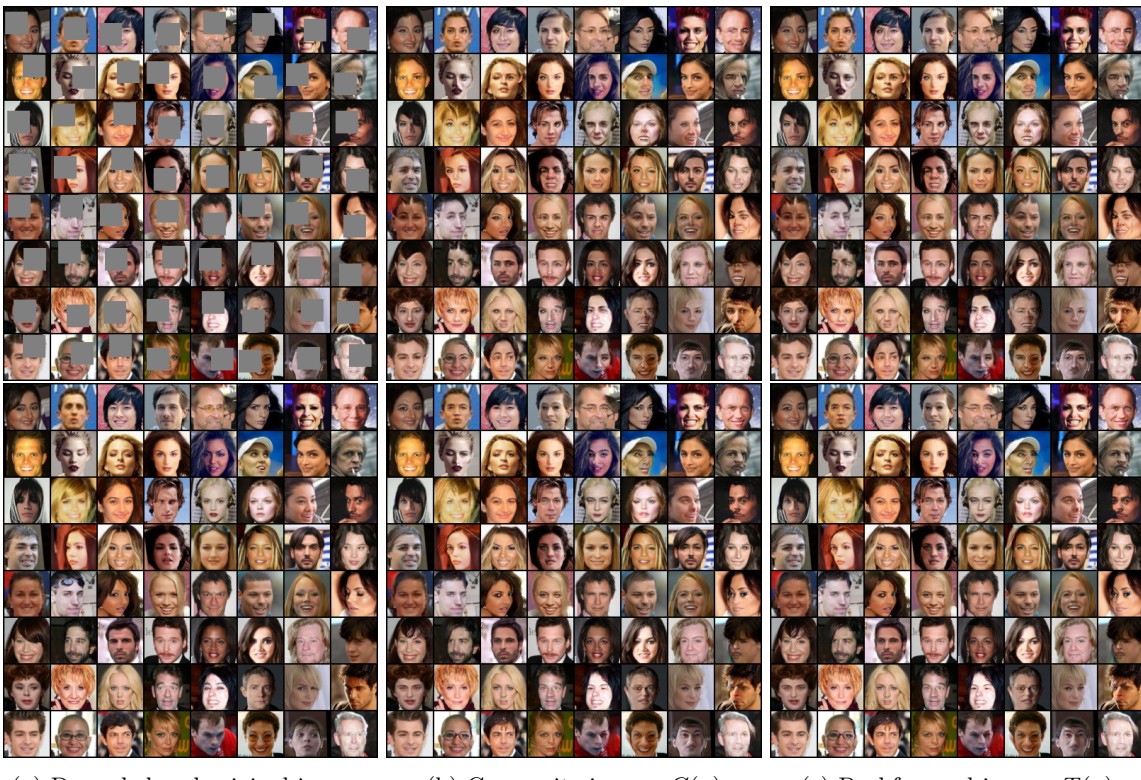

(a) Degraded and original images     (b) Composite images $G(x)$     (c) Pushforward images $T(x)$

Figure 14: Unpaired image inpainting on **test** dataset of CelebA $64 \times 64$. We take the composite image $G(x) = T(x) \odot M^C + x \odot M$ as the output image. Additionally, we provide the pushforward images $T(x)$ to illustrate the regularization effect of transportation costs. In panels (b) and (c), we show the results with $\alpha = 10$ in the first row and $\alpha = 10000$ in the second row. A small transportation cost would result that the pushforward map neglects the connection to the unmasked area, which is illustrated by a more clear mask border in pushforward images.

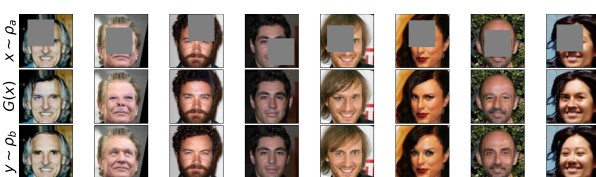

Figure 15: Unpaired image inpainting on **test** dataset of CelebA $128 \times 128$. We take the composite image $G(x) = T(x) \odot M^C + x \odot M$ as the output image.

We download CC-3M following this instruction. We then remove all the images with watermark, which include images downloaded from *shutterstock, alamy, gettyimages,* and *dailymail.co.uk* websites. After removing watermark images, the CC-3M only has 0.8M (text, image) pairs left. We then use CLIP retrieval and embedding-dataset-reordering packages to convert raw images/texts to embeddings.

For each dataset, we let the source and the target distribution contain 0.3M data respectively, and take the rest of dataset as the test data.

**Network structure and hyper-parameters** We use the DALLE·2 prior network to represent the map $T$. The potential $f$ is using the same network with an additional pooling layer. We use EMA to stabilize the training of the map. The batch size is 225. The numbers of loop iterations are $K_1 = 10$, $K_2 = 1$. We

use the learning rates $10^{-4}$, Adam (Kingma & Ba, 2014) optimizer with weight decay coefficient 0.0602. We train the networks for 110 epochs.

On NVIDIA RTX A6000 (48GB), the training time of each experiment is 21 hours.

### E.3    Unpaired inpainting

Denote $M^C$ as the complement of $M$, i.e. $M^C = 1$ in the occluded area and 0 otherwise. The composite image is $G(x) = T(x) \odot M^C + x \odot M$. The loss function we actually use is slightly different with the (5). We modify the $f(T(x))$ to be $f(G(x))$ to strengthen the training of $f$

$$\sup_f \inf_T \int_{\mathbb{R}^n} \left[ c(x, T(x)) - f(G(x)) \right] \rho_a(x) \, dx + \int_{\mathbb{R}^m} f(y) \rho_b(y) \, dy.$$

In the unpaired inpainting experiments, the images are first cropped at the center with size 140 and then resized to $64 \times 64$ or $128 \times 128$. We choose learning rate to be $1 \cdot 10^{-3}$, Adam optimizer with default beta parameters, $K_2 = 1$. The batch size is 64 for CelebA64 and 16 for CelebA128. The number of inner loop iteration $K_1 = 5$ for CelebA64 and $K_1 = 10$ for CelebA128.

We use exactly the same UNet for the map $T$ and convolutional neural network for $f$ as Rout et al. (2022, Table 9) for CelebA64 and add one additional convolutional block in $f$ network for CelebA128. On NVIDIA RTX A6000 (48GB), the training time of CelebA64 experiment is 10 hours and the time of CelebA128 is 45 hours.

We use the POT implementation (Flamary et al., 2021) of Perrot et al. (2016) as the template and implement the masked MSE loss by ourselves. Perrot et al. (2016) provides two options as the transformation map, one is linear and another is the kernel function. We find the kernel function on this example would generate mode-collapse results, so we adopt the linear transformation map. We choose $L_2$ regularization coefficient to be $10^{-3}$, and other parameters are the same as default.

### E.4    Class preserving mapping

The NIST images are rescaled to size $32 \times 32$ and repeated to 3 channels.

We use the conditional UNet to represent map $T$. We add a projection module (Miyato & Koyama, 2018) on WGAN-QC's ResNet (Liu et al., 2019) to represent potential $f$. We choose learning rate to be $1 \cdot 10^{-4}$, Adam optimizer with betas $(0.5, 0.999)$. $K_1 = 10, K_2 = 1$. We use EMA to stablize the training of the map. The batch size is 64. In practice, we introduce a coefficient $\lambda$ in the cost $c(\{x, y\}, \{x', y'\}) = \|x - x'\|^2 + \lambda \mathbf{1}(y \neq y')$ and set $\lambda = 0.5$.

# F   Theoretical justification of all the examples considered in this article

Table 4:   We make a summary table on all the problems considered in this work with their cost function $c$ and distributions $\rho_a, \rho_b$ and whether the existence and uniqueness of Monge map for each problem are guaranteed. We conclude the last column by checking the conditions in Theorem 1. Note that the theory of the existence of Monge map relies on the connection to a deterministic OT plan (Santambrogio, 2015, Section 1.1-1.4), and it is quite common that the OT plan is non-deterministic in deep learning examples. $\mathbb{S}^d$ is the unit ball in $\mathbb{R}^d$, and $\Delta^{d-1}$ is the $d-1$ dimension probability simplex in $\mathbb{R}^d$.

| Problem | $\mathcal{X}$ | $\mathcal{Y}$ | Cost | Guarantee of existence and uniqueness |
|---|---|---|---|---|
| Text to image | $\mathbb{R}^{77\times768} \setminus \{0\}$ | $\mathbb{S}^{768}$ | $c(x,y) = -\frac{\langle Rx,y\rangle}{\|Rx\|_2}$ | [No] $\nabla_x c(x,\cdot) = \frac{R^\top}{\|Rx\|}(I - uu^\top)$ may not be injective. Here we denote $u = \frac{Rx}{\|Rx\|}$. |
| Class-preserving map | $[-1,1]^{32\times32\times3} \times \Delta^{J-1}$ | $\mathcal{Y} = \mathcal{X}$ | $c(\{x,y\}, \{\bar{x}, \bar{y}\}) =$ $\|x - \bar{x}\|^2 - \lambda y^\top \bar{y}$ | [No] Consider $M = \mathbb{R}^{32\times32\times3+J}$, $\rho_a$ is not absolutely continuous as a distribution on $\mathbb{R}^{32\times32\times3+J}$ |
| Image inpainting | $[-1,1]^{32\times32\times3}$ or $[-1,1]^{64\times64\times3}$ | $\mathcal{Y} = \mathcal{X}$ | $c(x,y) \propto \|x \odot M - y \odot M\|_2^2$ | Existence guaranteed uniqueness not guaranteed |
| Population transport | $[0,2\pi) \times [0,\pi]$ | $\mathcal{Y} = \mathcal{X}$ | $c(\{\theta_1,\phi_1\}, \{\theta_2,\phi_2\}) =$ $-\sin\phi_1\sin\phi_2\cos(\theta_1-\theta_2)$ $-\cos\phi_1\cos\phi_2$ | [No] $\nabla_x c(x,\cdot)$ may not be injective, c.f. Remark 7 |
| Line to ellipse | $[-L,L]$ | $(0,\pi)$ | $c(x,y) =$ $(x - a\cos y)^2 + b^2\sin^2 y$ The parametrization of the line and the ellipse are $(x,0)$; $(a\cos y, b\sin y)$. | [Yes] The conditions in Theorem 1 are satisfied. |
| Annulus (reciprocal cost) | $\Omega_1$ $4 \le r \le 6$ | $\Omega_2$ $1 \le r \le 2$ | $c(x,y) = \frac{1}{\|x-y\|^2}$ | [Yes] One can restrict $c$ on $\mathcal{X} \times \mathcal{Y}$, i.e., set the manifold $M = \Omega_1$ as stated in Appendix A, and choose $\mathcal{X} = M$. The conditions in Theorem 1 are all satisfied. |
| Annulus ($L^2$ cost) | $\Omega_1$ $4 \le r \le 6$ | $\Omega_2$ $1 \le r \le 2$ | $c(x,y) = \|x - y\|^2$ | [Yes] One can verify that the conditions in Thm 1 are satisfied. |
| OT on sphere | $[0,2\pi) \times [0,\pi]$ | $\mathcal{Y} = \mathcal{X}$ | $c_\lambda(\{\theta_1,\phi_1\}, \{\theta_2,\phi_2\}) =$ $\arccos(\lambda(\cos\phi_1\cos\phi_2$ $+\sin\phi_1\sin\phi_2\cos(\theta_1-\theta_2)))$ | [No] $\nabla_x c(x,\cdot)$ may not be injective. c.f. Remark 7 |

**Remark 7.** *We discuss the condition* (18) *for geodesic cost. Consider the cost $c$ is the linearized geodesic distance* (14), *one can compute*

$$\nabla_{\theta_1,\phi_1} c(\{\theta_1,\phi_1\}, \{\theta_2,\phi_2\}) = \begin{pmatrix} \sin\phi_1\sin\phi_2\sin(\theta_1-\theta_2) \\ -\cos\phi_1\sin\phi_2\cos(\theta_1-\theta_2) + \sin\phi_1\cos\phi_2 \end{pmatrix}$$

*Fix $\theta_2 = \theta_1$. When $\phi_1 \neq \frac{\pi}{2}$, we can always choose two different $\phi', \phi'' \in [0,\pi]$ such that $\sin(\phi_1 - \phi') = \sin(\phi_1 - \phi'')$, so*

$$\nabla_{\theta_1,\phi_1} c(\{\theta_1,\phi_1\}, \{\theta_2,\phi'\}) = \begin{pmatrix} 0 \\ \sin(\phi_1-\phi') \end{pmatrix} = \begin{pmatrix} 0 \\ \sin(\phi_1-\phi'') \end{pmatrix} = \nabla_{\phi_1} c(\{\theta_1,\phi_1\}, \{\theta_2,\phi''\}),$$

*thus $\nabla_x c(x,\cdot)$ is not injective on $[0,2\pi)\times[0,\pi]$. One can also use a similar way to show that for the cost $c_\lambda$ (cf. (54)) used in OT on sphere, $\nabla_x c_\lambda(x,\cdot)$ is not injective. This is also true for the original arc length distance function (13). The cost function $c(x,y)$ may satisfy (18) if it is the square of the geodesic distance, i.e., $c(x,y)=d^2(x,y)$ where $d(x,y)$ denotes the length of the geodesic joining $x,y$. The computation w.r.t. the square cost may serve as one of our future research directions.*

## G    Notations used in the paper

| Notation | Meaning |
|:---:|:---:|
| $c(x,y)$ | cost function |
| $\mathcal{X},\mathcal{Y}$ | Polish spaces |
| $\rho_a,\rho_b$ | Borel probability measures defined on $\mathcal{X},\mathcal{Y}$ |
| $T$ | Measurable map from $\mathcal{X}$ to $\mathcal{Y}$ |
| $T_\sharp\mu$ | Pushforward of distribution $\mu$ by the map $T$ |
| $\mathcal{M}(\mathcal{X},\mathcal{Y})$ | Space of measurable maps from $\mathcal{X}$ to $\mathcal{Y}$ |
| $C_b(\mathcal{X})$ | Space of bounded continuous functions defined on $\mathcal{X}$ |
| $\mathscr{P}(\mathcal{X})$ | Space of Borel probability distributions defined on $\mathcal{X}$ |
| $C(\rho_a,\rho_b)$ | OT distance from $\rho_a$ to $\rho_b$, i.e., optimal value to the general OT problem 2 |
| $C_{\mathrm{Monge}}(\rho_a,\rho_b)$ | Optimal value to the Monge problem 1 |
| $K(\rho_a,\rho_b)$ | Optimal value to the Kantorovich dual problem (3), (19), or (20) |
| $\mathcal{L}(T,f)$ | Lagrangian function of the Monge problem, defined in (5) |
| $\psi^{c,+}$ | $c-$transform defined in (21) |
| $\phi^{c,-}$ | $c-$transform defined in (22) |
| $T_*,\phi_*$ | Optimal solution (Monge map) to (1) and optimal solution to (20) |
| $(\hat{T},\hat{f})$ | Saddle point of $\mathcal{L}(T,f)$ to (4) |
| $(\bar{T},\bar{f})$ | sup-inf solution to (4) |
| $B,K,K_1,K_2$ | Training batch size, total iterations, iterations for $T_\theta$, iterations for $f_\eta$ |
| $R$ | Frozen matrix |
| $\odot$ | Point-wise multiplication |
| $M,M^C$ | Binary mask with the same size as the image, complement of $M$ |
| $U(E)$ | Uniform probability distribution on measurable set $E$ |

