# OpenReview forum: "Neural Monge Map estimation and its applications"
_TMLR — Accepted by TMLR_

### Review · Reviewer_LY47 · 2023-02-07

**Summary Of Contributions:**

This paper proposes a method to compute the Monge map (modeled by a neural network) for the optimal transport problem. The proposed method is simply based on the Lagrange multiplier method applied to the Monge problem, which induces a saddle point problem, models the Monge map and multipliers with neural networks, and does not rely on any regularization techniques that are often involved in the recent large-scale optimal transport methods. This formulation has the advantages such as (1) the ability to obtain the Monge map, (2) being an accurate estimate of the optimal transport distance because it is free from regularizers, (3) being more scalable than the previous neural-network-based optimal transport methods because the architecture is not restricted to a specific one, and (4) the applicability to broader cost functions not only restricted to the L2 ground cost. Thanks to these benefits, the authors showed fantastic applications such as the unpaired text to image generation (taking the cosine similarity between texts and images as the ground cost), class-preserving maps (taking the distance defined on x and y jointly as the ground cost), unpaired image inpainting (taking the MSE in the unmasked areas as the ground cost), and population transportation estimation (taking the geodesic distance as the ground cost).

**Audience:**

Yes

**Claims And Evidence:**

Yes

**Requested Changes:**

### Suggestions and questions

- In Section 3, some technical details are deferred to the appendix, and eventually a lot of equations are referred without their formal explanations in the main text. For example, "the dual problem (19)" in Theorem 2 is an important concept but not mentioned in Section 3. Can you have a bit more words to explain it here?
- In Section 3 "Mapping between unequal dimensions," the authors claim that the proposed method can be used even when the dimensions of the source and target space are different. However, I suppose that this is a natural consequence of the usual optimal transport problem when we only manipulate these two spaces via the corresponding cost function. Is it specific to the proposed method?
- In Section 6.3, the authors claim "most real-world applications do not involve the paired datasets," but in common scenarios of self-supervised learning such as the masked autoencoders, I think it is pretty natural to mask an input and reconstruct the original data from the occluded one, where the pair data are naturally available. Does the author consider other setups?
- In Section 7, the authors claim that the proposed method is more advantageous than other deep learning methods for being free from paired data, but some unpaired experiments in Section 6 may not be sufficiently natural. For example, the text-to-image generation (Section 6.1) uses the pretrained CLIP model, which requires paired data to fit. The experiments in Section 6.3 may be natural to have paired data (as mentioned above). Can you discuss and be upfront if necessary in this perspective?

### Minor comments

- In the header of Theorem 1, "Existence" is better to be in lowercase.
- In Theorem 1, you have the notation $(Id, T\_*)$, which is not introduced formally and inconsistent with $[\mathrm{id}, T_\*]$ right before Section 6.
- In Assumptions 3 and 4, "argmin" should be denoted by `\mathrm`. In addition, "arginf" may not make sense because the infimum does not necessarily suppose the existence of the minimizer. I think "argmin" is sufficient for the purpose.
- In p.11, the authors cited Fréchet Inception Distance for the first time, but it appears in p.10 (as FID). It seems better to cite there.

**Strengths And Weaknesses:**

## Strengths

1. The proposed method is based on a simple idea, solving the Monge problem by the Lagrange multiplier method. This does not require any regularizers and architectural constraints that degenerate the distance estimation accuracy and numerical stability.
2. The proposed method does not need to deal with the "joint" constraints in the Kantorovich dual $\\phi(y) - \\psi(x) \\le c(x, y)$, which is an important component to make optimization independent from the paired data $(x, y)$, being scalable. Despite that a closely-related work Makkuva et al. (2020) escapes from this constraint by using input-convex neural networks, this restricts our architectural design and hinders the scalability. The proposed method does not have this constraint.
3. The authors evaluated the effectiveness of the proposed method with a number of applications (as I mentioned above). In particular, I found the application to text-to-image generation is really fascinating one. This is sufficiently persuasive that the proposed method is scalable to the high-dimensional setup.

## Weaknesses

1. The estimation error analysis in Theorem 4 is rather complicated and involves the weight function $\\beta$ to measure the estimation error. This is an additional artifact to the previous error analysis given by Makkuva et al. (2020). Further, by looking at the proof, I found that the weight function depends on the transport map $T$ itself (because the definition of the weight function in Eq. (42) involves $\\omega$, which depends on $T$ in Eq. (37)). May I ask you to discuss why the weight function is needed? In addition, are there any issues of the weight function being dependent on the transport map?
2. The authors claim that "for large scale problems ICNN is not expressive enough" in Section 5, and advocates for the proposed method. Is it possible to support this claim experimentally? I don't intend to argue that the current experiments are insufficient (indeed, the experiments given by the authors are very thorough, in my opinion!), but simply would like to highlight the advantage of the proposed method.

---

> ### Author Response · Authors · 2023-04-09
> **Reviewer LY47 Response (1/2)**
>
> We thank the reviewer for this thorough review and for the kind words about the quality of our work. Below we address the specific questions and comments.
>
> ## Theorem 4: weight function
>
> > May I ask you to discuss why the weight function is needed? In addition, are there any issues of the weight function being dependent on the transport map?
> >
>
> You are completely right here. In our result, the weight function $\beta$ actually depends on both the current computed map $T$ and the dual variable $f$.
>
> We should first clarify that the coefficient $\alpha$ introduced in [Makkuva] is not a ``real" constant, it is the minimum eigenvalue of the Hessian of the dual $f$ in the algorithm. Actually, our theoretical result is a generalization of [Makkuva], this is stated in Remark 1 after Theorem 4 in our original paper. By taking a minimum of $\beta$, we can also move the weight function outside of the $L^2$ loss, but the inequality we obtained will be looser compared with the original one. So, we decide to leave the weight $\beta$ in our analysis result.
>
> We also emphasize that we are establishing an a posteriori error estimation. This means that we are able to provide an error upper bound only when we have computed for certain $(T, f)$. Since $(T, f)$ could be arbitrary, we cannot anticipate the weight $\beta$ to be uniformly bounded from below. This indicates that it is possible that for certain $(T, f)$, our error upper bound is large and may fail to quantify the closeness between $T$ and $T_*$. Furthermore, Example 2 proposed at the end of Appendix B.5 indicates that there do exist certain OT problems whose corresponding weight function $\beta$ is always small, even though $T$ is very close to the Monge map $T_*$.
>
> To conclude, in this paper we provide a posterior upper bound for the $L^2$ error between computed $T$ and Monge map $T_*$ in Theorem 4. But we do have no guarantee on uniformly small upper bound since the weight $\beta$ will rely on the pair $(T,f)$ which could be arbitrary.
>
> ## Theorem 2
>
> > the dual problem (19)" in Theorem 2 is an important concept but not mentioned in Section 3
> >
>
> We agree with the reviewer on the importance of the Kantorovich problem (19). We have mentioned its importance and relation with our proposed max-min problem right after the introduction of (4) in our revised paper.

---

> ### Author Response · Authors · 2023-04-09
> **Reviewer LY47 Response (2/2)**
>
> ## Other statements
>
> > The authors claim that "for large scale problems ICNN is not expressive enough" in Section 5, and advocate for the proposed method. Is it possible to support this claim experimentally
> >
>
> This claim is not trivial, and it is supported by several previous works.
>
> Firstly, Korotin et al. (2021a) demonstrated that ICNN-based methods are not suitable for large-scale problems. They used ICNN-based solvers and our method with quadratic cost (or [MM:R]) to estimate the loss for GANs, and their experiment showed that [MM:R] outperformed ICNN-based solvers by a large margin in image generation tasks. Specifically, their Figure 5a (ICNN-based solver) and Figure 5d ([MM:R]) show the comparison, with [MM:R] achieving an FID score of 18.8, while the ICNN-based solver only achieves a score of 90.
>
> Fan et al. (2022b) also showed that regular (non-ICNN) parameterization leads to notably improved performance in an OT-related task. Their Figure 3a and 4a provide evidence for this claim.
>
> More recently, Korotin et al. (2022a) used our method with quadratic cost (or [MM:R]) to solve the Wasserstein barycenter in CelebA image pixel space. They compared with another Wasserstein barycenter baseline (Fan et al., 2020) with ICNN parameterization and found that using [MM:R] outperformed the ICNN-based baseline. Their Figure 6 provides a visual comparison.
>
> > Mapping between unequal dimensions
> >
>
> Given the cost $c(x,y) = \bar c(Q(x), y)$ we discussed, it could be possible that the usual OT methods between the space of $Q(x)$ and $y$ have the same effect with our method (see Lemma 4.2 of Rout et al., 2022). However, during the inference, applying our learned map offers the advantage of directly mapping the data $x$ without the need for an intermediate step of computing $Q(x)$, potentially streamlining the process.
>
> On the other hand, in some cases, a more intricate cost function may be necessary, where the cost cannot be decomposed as a function of only $Q(x)$ and $y$. In such scenarios, applying traditional OT methods directly could become challenging. The creation of such a cost function may be a complex task, which we could consider exploring in future research.
>
> > Unpaired datasets can be from semi-supervised methods
> >
>
> We appreciate your suggestion and have made the modification to our statement accordingly.  We change our phrase to *“**some** real-world applications do not involve the paired datasets.”* However, it is worth noting that even in scenarios where a self-supervised learning approach can be taken, it often requires additional pre-processing efforts. On the other hand, it is possible that some applications do not involve paired datasets. For instance, the source data may consist of degraded images of old faces while the target data contains intact young faces. In such scenarios, there is a distributional shift between the two datasets that cannot be simply addressed by occluding young faces to construct the input data.
>
> > Paired data claim in Conclusion
> >
>
> We have weakened our claim to be “our scheme reduces the need for paired data”.
>
> > Minor comments
> >
>
> Thank you for your comments. We have modified the paper accordingly.
>
> Specifically, the notation $(\textrm{Id}, T_*)_\sharp\rho_a$ is explained in the footnote after Theorem 1. This notation is used there for the first time.

---

### Review · Reviewer_u8z1 · 2023-02-13

**Summary Of Contributions:**

In this articles, authors propose to compute the Monge map and the associate transport cost using a general cost. They reformulate the Monge problem using the Lagrangian and derive conditions for which having solutions to the sup-inf problem on the Lagrangian is equivalent with having a Monge map. Then, they propose to solve this problem in practice by modeling the transport map and the dual potential with neural networks, and by solving the sup-inf problem with the Lagrangian as objective. This objective can be approximated provided samples from the source and the target distributions.

On the theoretical side, they provide a duality gap result. On the experimental side, they provide results for different problems using different ground cost, and hence demonstrating the versatility of their method.

**Audience:**

Yes

**Claims And Evidence:**

No

**Requested Changes:**

I have some quesions about the experiments, in particular on what they aim to show. If I understand well, these experiments aim at showing that this method is well able to approximate the OT map and the Wasserstein distance for general costs in different large scale settings. But no comparison with a ground truth is provided. Hence, I believe that an experiment comparing the map obtained and the value of the objective with a groundtruth Monge map and Wasserstein distance is missing. A possibility would be to use the benchmark introduced in [1] for 2-Wasserstein. However, since the proposed method coincides with other methods in this case, and hence has already been benchmarked, another toy case might also be of interest. Closed-form for other costs than the quadratic cost might be intricate to find. Maybe a possiblity would be to use the construction of [2] to approximate a c-concave map on the sphere, and hence build a Monge map for the Wasserstein distance on the sphere with geodesic cost. Another possibility between discrete samples might be on the manifold of SPDs with log-euclidean metric as in Theorem 1 of [3].
Then, I am not always convinced with the baseline used in the experiments. For example, in the unpaired text to image generation, it is written "We use the same unpaired data generation scheme as [4]" but it is not compared with [4]. Providing such comparison would allow to further assess the superiority of using the negative cosine similarity as cost over the quadratic distance.
In the experiment on the sphere, the comparison is performed with [5] which is not really comparable since it uses a linear transformation. I believe that an additional comparison with the groundtruth plan obtained e.g. with Sinkhorn as in [6] would allow to better appreciate the map. And instead of [5], a method using NN with 2-Wasserstein might be more comparable, even though we expect that it would not satisfy the spherical geometry.

[1] Korotin, Alexander, et al. "Do neural optimal transport solvers work? a continuous wasserstein-2 benchmark." Advances in Neural Information Processing Systems 34 (2021): 14593-14605.

[2] Cohen, Samuel, Brandon Amos, and Yaron Lipman. "Riemannian convex potential maps." International Conference on Machine Learning. PMLR, 2021.

[3] Ju, Ce, and Cuntai Guan. "Deep Optimal Transport on SPD Manifolds for Domain Adaptation." arXiv preprint arXiv:2201.05745 (2022).

[4] Rout, Litu, Alexander Korotin, and Evgeny Burnaev. "Generative modeling with optimal transport maps." arXiv preprint arXiv:2110.02999 (2021).

[5] Perrot, Michaël, et al. "Mapping estimation for discrete optimal transport." Advances in Neural Information Processing Systems 29 (2016).

[6] Amos, Brandon, et al. "Meta optimal transport." arXiv preprint arXiv:2206.05262 (2022).

**Strengths And Weaknesses:**

The paper is clearly written and easy to follow. The method proposes to approximate the Monge map for general costs by solving the sup-inf problem on the Lagrangian. While it has already be done for the quadratic cost, the method seems novel for more general costs. Moreover, experiments demonstrate well that the method works for different costs and in large scale settings.

Strengths:
- A method to solve the Monge problem for general costs
- Theoretical results with duality gap
- Several experiments demonstrating that the methods works in large scale settings

Weaknesses:
- It lacks a quantitative study of the quality of the Monge map learned, and of the Wasserstein distance approximated
- For some experiments, the superiority of using the proposed cost over the classical quadratic cost is not shown (e.g. in Unpaired text to image generation)

---

> ### Author Response · Authors · 2023-04-09
> **Reviewer u8z1 Response**
>
> We thank the reviewer for the useful and in-depth feedback on our work. Below we address the reviewer’s concerns and questions:
>
> ## Lack of a quantitative study
>
> The quality of the Monge map can be evaluated from two criteria: 1) Whether the constraint  $T_\sharp \rho_a \approx \rho_b$ is satisfied? We have verified this result both quantitatively and qualitatively. Quantitative results show up in Figure 4, Table 1 and 2. (2) Whether the objective function can recover Wasserstein distance.
>
> As the reviewer has pointed out, the 2-Wasserstein distance has been benchmarked by Korotin et al. (2021a) and Rout et al., (2022). **As for the toy case of quadratic cost, we will refer readers to 2D example comparisons in Figure 16 and Figure 17 of Rout et al., (2022).** Specifically, they show this method can outperform LSOT (Seguy et al. 2018), WGAN-LP, and W2GN (Korotin et al., 2021b) and is on par with ICNN-OT (Makkuva et al., 2020). Due to respect to their work, we think their comparison is reliable and thorough, therefore it is not necessary to redo the comparisons. For the **non-quadratic** cost, please see the next bullet point.
>
> ## Comparison with ground truth Monge map for a non-quadratic cost
>
> **In the appendix, we do have the comparisons with the ground truth Monge map with non-quadratic cost. The examples in Figure 9 and Figure 10 both have closed-form solutions. We have calculated the Wasserstein distance and objective function value for them, which are presented in Table 3.**
>
> Regarding the two possibilities proposed by the reviewer, the first one (Cohen et al. 2021) suggests a more delicate way to parameterize the Monge map based on the theory of McCann (2001) "Polar factorization of maps on Riemannian manifolds. Geometric & Functional Analysis GAFA". However, it cannot be used as a ground truth map because its generated distribution shows visible discrepancies with the target, as demonstrated in their Figure 3 checkerboard example and Figure 4. We also want to comment that their training process requires access to the closed-form density of the source and/or target distribution, whereas our algorithm only requires samples from both distributions. So their method can not be directly applied to the population transport example in our Sec 6.4. We appreciate the reviewer's suggestion and have added these two papers to the literature.
>
> ## Comparison with the quadratic cost for text-to-image generation
>
> We also compared the cos-sim cost (9) and the ordinary inner product cost c(x, y) = −⟨Rx, y⟩. The latter can converge to the same level as cos-sim cost, measured both qualitatively via generated images and quantitatively via cosine similarity w.r.t. real images (see Figure 4 (b).) The only difference is the method with inner product cost will converge slightly slower. After investigating the statistics of ∥Rx∥, we find this result is reasonable because ∥Rx∥ concentrates around 12 with a high probability. Meanwhile, we emphasize that this text-to-image generation task has not been considered in previous neural OT methods, which is itself a contribution, even though using cos-sim cost does not distinguish itself from inner product cost significantly.
>
> We have included this discussion at the end of Section D.
>
> ## Transportation on the sphere
>
> We first point out that in our setup, the target distribution is a uniform distribution, so the Sinkhorn algorithm can only give a transport plan but not a map. So we can not directly apply the Sinkhorn algorithm if we assume the mass can not be split, e.g. each particle represents a person. On the other hand, Amos et al. (2022) designs the target to be a discrete measure, so the Sinkhorn algorithm can give a valid map. We have tried our method on the same setup as Amos et al. (2022) but unfortunately, our method tends to be rather unstable in this setting. We summarized this problem in the *Limitation* paragraph.
>
> **We have added the comparison between the approximated geodesic $c_\lambda$ cost (defined in eq. (54)) and the quadratic $L^2$ cost in Figure 12.** **Explanations of the comparison are presented at the end of the subsection “Population transportation” of Appendix D.**

---

### Review · Reviewer_HSqP · 2023-03-27

**Summary Of Contributions:**

In this paper, the authors propose a framework for estimating the Monge map that is parameterized as neural networks between probability measures of large-scale datasets. To derive algorithms that are practically feasible, the authors first reformulate the Kantorovich dual of OT into a maximin saddle point problem. Specifically, the authors provided equivalence and consistency theoretical results to connect the proposed saddle point objective with the established results regarding the existence and uniqueness of Monge maps. Thus, the optimal solution to the saddle point objective corresponds well to optimal map estimation. From an algorithmic perspective, the authors parameterized both the transport map and the dual potential as neural networks to obtain a practical algorithm for training a neural Monge map. The proposed method can employ different kinds of ground cost metrics. Moreover, it can transport probability measures of different dimensions using predefined embedding functions.

From a theoretical perspective, the authors provided a detailed analysis regarding the duality gap between the estimated map and an optimal Monge map. The $L^2$ error is guaranteed to be upper-bounded. The authors have presented a list of experimental results to validate the effectiveness of their proposed method. Specifically, the authors have used multiple cost metrics for different data tasks, including large-scale unlabeled text-to-image transfer, label-conditioned transfer, image in-painting, and population transport.

**Audience:**

Yes

**Broader Impact Concerns:**

It appears that the authors have not discussed the broader impact of their work in the current version of the draft. In my opinion, the proposed method can serve as a powerful generative model, but there may be associated risks. On the other hand, using optimal transport in the formulation provides more interpretability. Therefore, I encourage the authors to include some discussion on the broader impact of their work in the draft.

**Claims And Evidence:**

Yes

**Requested Changes:**

Overall, I think this draft has satisfactory quality, and I recommend acceptance particularly if the points in the weaknesses are addressed:

- When adding more discussions regarding the smoothness regularization of the optimal transport map estimation, the authors may want to refer to Theorem 3 of Manole et al. (2021).
- More about the advantages and disadvantages between deterministic maps and stochastic maps.
- Could you provide more clarification on the text-to-image generation experimental results?
- How are the label correspondences selected in the class-preserving map?

[1] Manole, T., Balakrishnan, S., Niles-Weed, J., & Wasserman, L. (2021). Plugin estimation of smooth optimal transport maps. *arXiv preprint arXiv:2107.12364*

**Strengths And Weaknesses:**

Strength:

- Firstly, this paper is well-written, with clear motivation and explanation, along with sufficient experimental evaluation. The organization of the contents is also good.
- The motivation is strong, as there is clearly a need to improve the understanding of OT map estimation in light of the emergence of deep-learning-based large models. In particular, the example of estimating transport without text-image pair labels will be very useful given the fact that large language models are being deployed in various ML areas.
- The theoretical results are well-presented. Although the idea of transforming the Kantorovich dual into the saddle point has appeared in existing studies, the approach to understanding the duality gap is quite novel.

Weakness:

- Questions regarding theoretical results:
    - The theoretical results in this work are established on Theorem 1. I would like to learn more about the regularization conditions for the Monge map itself. Specifically, when estimating the transport map between two distributions, the existence and uniqueness of the optimal transport map will require a certain degree of smoothness (more than that of the source and target measure). Should we also include regularization for the map in this work? What could be the effect of regularization, and is it possible to do it for a Monge map that is parameterized as a neural network?
- Questions regarding experimental results:
    - In section 6.1, the authors displayed qualitative results on the Laion Art dataset and the CC-3M dataset, respectively. Since the CC-3M dataset does not have text-image-pair labels, how were the "real image" and the "real embedding" generated in Figure 2?
    - In Figure 4, is it possible to show the pushforward embedding distribution heat-map obtained by some baseline OT methods such as the linear map [Perrot et al]?
    - In Figures 1 and 2, it seems that the decoded images from the proposed method have better fidelity than those from the diffusion prior. What could be the cause? Is it possible to elaborate more on the advantages and disadvantages between a deterministic map and a stochastic map?
    - The authors could provide more information regarding the label-preserving map. How is the correspondence between labels selected?

---

> ### Author Response · Authors · 2023-04-09
> **Reviewer HSqP Response: Questions regarding theoretical results**
>
> We thank the reviewer for the thoughtful review and for the kind words about the quality of our work. Below we address the reviewer’s specific questions and comments:
>
> > Should we also include regularization for the map in this work? What could be the effect of regularization,
> >
>
> We thank the reviewer for reminding us about the regularity (regularization) issue of the Monge map. Based on the references mentioned by the reviewer, we think the review actually means “regularity” instead of “regularization” here. The regularization term on enforcing $\phi(x)-\psi(y)\leq c(x,y)$ has been automatically built in the inner minimization process so we do not have to further add it to our max-min optimization scheme.
>
> The regularity of the Monge map does play an important role in Optimal Transport (OT). It is also a crucial topic related to the Monge-Ampere equation associated with the OT problem. Under our setting, in Theorem 1, the c-convex dual $\psi_*$ do possess a basic regularity property, that is $\psi_*$ is differentiable, thus we are able to take the gradient of it. Such regularity result is a direct corollary of Theorem 10.26 from [Villani (2008)]. The missing details on explaining this regularity result have been added to Theorem 1 and Remark 3 in our revised paper. Furthermore, the reviewer is more interested in stronger regularity such as $C^{k,\alpha}$ smoothness. There do exist regularity results for general cost function $c(x,y)$. One may refer to Theorem 12.52 of [Villani (2008)] for more details on $C^{k,\alpha}$ regularity of the optimal dual variable $\psi_*$. It is definitely true that regularity plays an important role in analyzing the proposed algorithms for OT. As mentioned by the reviewer, in [1], the authors are focusing on analyzing the minimax rate of certain types of estimators under the statistical setting, which is going beyond what we have done in our paper—We did not consider the statistical setting and directly treat both the source and the target distributions as exact probability measures instead of empirical measures. As a result, in order to obtain the theoretical bounds in [1], one may need more sophisticated properties on $\psi_*$ such as the $C^{k,\alpha}$ smoothness. We carefully checked our arguments and found that the further property such as $C^{k,\alpha}$ on $\psi_*$ is not really necessary under our problem setting. However, we do recognize the importance of stronger regularity on the Kantorovich dual function $\psi_*$. This may play a crucial role in our future research on analyzing the statistical behavior of our proposed algorithm. We have added a summary of the above discussion in Remark 5 of Appendix A of our revised paper.
>
> > and is it possible to do it for a Monge map that is parameterized as a neural network?
> >
>
> This is an interesting problem regarding the designing structure of neural networks. For networks with simple structure and smooth activation functions, such as Multilayer Perceptron, one can show that it is $C^\infty$ (i.e., smooth) on any bounded region, and any of its $C^{k, \alpha}$ norm should be bounded by finite number; similar results should hold for many more kinds of network structures. On the other hand, using a network with a non-smooth activation function such as ReLU, will not guarantee its regularity. The behavior and convergence of our algorithm with such a neural network remain as a challenging and important problem. It will serve as one future research direction.

---

> ### Author Response · Authors · 2023-04-09
> **Reviewer HSqP Response: Questions regarding experimental results**
>
> > How were the "real image" and the "real embedding" generated in Figure 2?
> >
>
> Indeed, cc-3m contains image-text pairs, which can be used in the test dataset. We follow the same procedures of *Laion art* to calculate the real embeddings. We first download images following this [instruction](https://github.com/rom1504/img2dataset/blob/main/dataset_examples/cc3m.md). We then run the *CLIP retrieval* to convert images to embeddings and use *embedding-dataset-reordering* to reorder the embeddings into the expected format. More details can be found in Sec E.2.
>
> > In Figure 4, is it possible to show the pushforward embedding distribution heat-map obtained by some baseline OT methods such as the linear map [Perrot et al]?
> >
>
> We have added this comparison in Figure 4(a).
>
> > In Figures 1 and 2, it seems that the decoded images from the proposed method have better fidelity than those from the diffusion prior. What could be the cause?
> >
>
> We observe that our method can generate reasonable images that are comparable to those generated by the diffusion prior, as demonstrated in Figure 1 using the Laion art dataset. Moreover, in Figure 2 using the CC-3M dataset, our images clearly outperform those generated by the DALL·E2-Laion model, which was trained on a different dataset and has never seen CC-3M before. We attribute this improvement to the fact that our OT map in Figure 2 is specifically trained on CC-3M, allowing it to generate more reasonable images in this scenario.
>
> > Advantages and disadvantages between a deterministic map and a stochastic map?
> >
>
> The loss function for the stochastic map incorporates the variance of the push-forward distribution, which encourages the generation of more diverse data. This can lead to lower FID scores when compared to the real target distribution. However, because the stochastic map requires random noise as input, it requires a more sophisticated network architecture and is more computationally expensive to train.
>
> > The authors could provide more information regarding the label-preserving map. How is the correspondence between labels selected?
> >
>
> In the label-preserving map, we assume that the correspondence between the labels in the source and target distributions is known or given, depending on the application scenario. For example, in the case of the MNIST and MNIST-M datasets, all gray images of digit 5 in the MNIST dataset correspond to colorful images of digit 5 in the MNIST-M dataset. In a biological application, this could correspond to control cells and perturbed cells by drugs, with the label representing the type of drug. In our paper, we deliberately chose to use datasets that do not have obvious correspondences, such as Fashion MNIST vs MNIST, in order to demonstrate the effectiveness of our method on more challenging tasks. By doing so, we can better highlight the advantages of our approach.
>
> > Could you provide more clarification on the text-to-image generation experimental results?
> >
>
> Figures 1 and 2 demonstrate that passing text embeddings directly into the image decoder can lead to dreadful results, highlighting the necessity of performing text-to-image alignment in the DALLE2 model. By comparing the *real embedding* and *real image*, we can assess the decoder recovery ability, which accounts for the fidelity gap between our generated images and the real images. The most relevant baselines for us are *prior embedding* and *real embedding*, and we show that our method can generate comparable images to those produced using real embeddings.
>
> Since our method maps text encoding to image embedding, it is essential that our generated embedding distribution matches the real embedding, even on the test dataset. Figure 4(a) qualitatively confirms that this is indeed the case. In Figure 4(b) (left plot), we quantitatively compare the cosine similarity between the generated embedding and the real embedding. A value of one indicates an exact match between the generated and real embeddings. In the right plot of Figure 4(b), we show the cosine similarity between the generated embedding and unrelated text embedding, with a lower value indicating better performance. A value of less than 0.1 is considered to be satisfactory.
>
> > I encourage the authors to include some discussion on the broader impact of their work in the draft.
> >
>
> We have summarized one paragraph about the broader impact at the end of the main paper.

---

> ### Comment · Reviewer_HSqP · 2023-04-24
> **Reply**
>
> I would like to express my gratitude to the authors for their effort in responding to my questions and making modifications to the manuscript. Here, I would like to provide further elaboration on the points that I raised:
>
> - Thank you for addressing most of my clarification problems in the updated manuscript. Specifically, the additional discussion on the regularity of the map would be very beneficial for readers of this work. It is an interesting but challenging future direction that requires more assumptions on the smoothness of performing analysis.
> - The additional comparison in Figure 4(a) is satisfactory, as it clearly demonstrates the advantage of the proposed method over existing map estimation methods. I think it is reasonable, especially for large-scale data, which is the focus of the authors' work.
> - I agree with the authors' discussion on the difference between deterministic maps and stochastic maps. Recent advances, such as OT map estimation [Liu et al 2022, rectified flow], have demonstrated the efficiency of estimating a generative model.
> - Thank you for providing more details about the label-preserving map and text-to-image generation. These have addressed my concerns.
>
> [Liu et al 2022, rectified flow] Flow Straight and Fast: Learning to Generate and Transfer Data with Rectified Flow

---

> > ### Author Response · Authors · 2023-04-25
> > **Reply by authors**
> >
> > Dear reviewer HSqP,
> >
> > Thank you very much for your reply! We are glad that your questions and concerns are addressed!
> >
> > Best regards,
> > Authors

---

### Decision · Action_Editors · 2023-06-06

**Recommendation:** Accept as is

**Comment:**

This paper proposes a Monge map estimation using neural networks. The authors first reformulate the Kantorovich dual of OT into a maximin saddle point problem. Then, they propose to solve this problem in practice by modeling the transport map and the dual potential with neural networks. Moreover, a detailed analysis regarding the duality gap between the estimated map and an optimal Monge map is provided. Through experiments, it demonstrates that the proposed method outperforms existing methods in various types of applications.

The paper works on estimating a Monge map, a fundamental problem in OT, and many OT researchers are interested in working on the topic. Specifically, they proposed a simple yet general framework based on the Lagrange multiplier method. This contribution reminds me of the work done by the work Cuturi 2013 (NIPS), which proposed an entropic regularized OT and proposed a Sinkhorn algorithm. Now, the Sinkhorn algorithm is used in many machine learning applications including document classification, matching, and divergence estimation. Since the contribution of this paper is simple and fundamental, it can be further studied like the Sinkhorn algorithm, and can be useful in many real applications.

Overall, the proposed neural Monge Map estimation approach is an important contribution to the OT community and the reviewers agree to accept the paper. In addition, one of the very experienced reviewers recommended a featured certification; I also vote for acceptance with a featured certification.



**Audience:**

Estimating a Monge map is one of the hot topics in optimal transport; OT researchers are in general interested in the work.

**Claims And Evidence:**

The claims are well addressed through experiments.